# LUCID: Attention with Preconditioned Representations

**Sai Surya Duvvuri** [* † 1]  **Nirmal Patel** [* 1]  **Nilesh Gupta** [† 1]  **Inderjit S. Dhillon** [2]

## Abstract

Softmax-based dot-product attention is a cornerstone of Transformer architectures, enabling remarkable capabilities such as in-context learning. However, as context lengths increase, a fundamental limitation of the softmax function emerges: it tends to diffuse probability mass to irrelevant tokens degrading performance in long-sequence scenarios. Furthermore, attempts to sharpen focus by lowering softmax temperature hinder learnability due to vanishing gradients. We introduce LUCID Attention, an architectural modification that applies a preconditioner to the attention probabilities. This preconditioner, derived from exponentiated key-key similarities, minimizes overlap between the keys in a Reproducing Kernel Hilbert Space, thus allowing the query to focus on important keys among large number of keys accurately with same computational complexity as standard attention. Additionally, LUCID's preconditioning-based approach to retrieval bypasses the need for low temperature and the learnability problems associated with it. We validate our approach by training $\sim$1 billion parameter language models evaluated on up to 128K tokens. Our results demonstrate significant gains on long-context retrieval tasks, specifically retrieval tasks from BABILong, RULER, SCROLLS and Long-Bench. For instance, LUCID achieves up to 18% improvement in BABILong and 14% improvement in RULER multi-needle performance compared to standard attention.

## 1. Introduction

Transformers, underpinned by the softmax dot-product attention mechanism, have become the cornerstone of modern

---

[*]Equal contribution †Work done during internship at Google.
[1]The University of Texas at Austin [2]Google. Correspondence to:
Sai Surya Duvvuri <saisurya@cs.utexas.edu>.

*Proceedings of the 43$^{rd}$ International Conference on Machine Learning*, Seoul, South Korea. PMLR 306, 2026. Copyright 2026 by the author(s).

machine learning, particularly in the domain of large language models (LLMs) (Vaswani et al., 2017). This mechanism, operating on queries (Q), keys (K), and values (V), allows models to dynamically weigh the importance of different parts of the input sequence. The exponential function within the softmax operation is crucial, enabling a sharp focus on relevant tokens, which forms the basis for remarkable capabilities such as in-context learning (Brown et al., 2020) and retrieval.

However, as the demand grows for LLMs to process increasingly longer sequences for tasks involving complex reasoning or extended documents, limitations of the standard attention mechanism become more apparent. Although softmax encourages focus, it often assigns non-negligible weight to irrelevant tokens, leading to diluted attention distributions. This phenomenon, often referred to as "attentional noise," can reduce precision and degrade performance on long-context tasks (Ye et al., 2025), and can hinder attention to scale gracefully with sequence length. Additionally, softmax's sensitivity to temperature can affect learnability of the model, high temperature can lead to representation collapse (Masarczyk et al., 2025) and low temperature can lead to saturation and training instabilities (Zhai et al., 2023).

Inspired by these developments, we ask whether such correction strategies can benefit standard softmax attention by targeting the root cause of the attention noise issue - correlated keys. Viewing softmax attention through the lens of kernel methods (Katharopoulos et al., 2020), where attention probability is proportional to an inner product in a Reproducing Kernel Hilbert Space (RKHS) $\exp(\langle \mathbf{q}, \mathbf{k} \rangle) = \langle \phi(\mathbf{q}), \phi(\mathbf{k}) \rangle$. The crucial insight at the core of our paper is that we decorrelate the keys in RKHS feature space by preconditioning $\phi(\mathbf{k})$. This allows the queries in the feature space $\phi(\mathbf{q})$ to focus on selected keys among large number of keys with minimal noise, improving the retrieval performance. Critically, LUCID bypasses the vanishing gradients associated with lowering softmax entropy to sharpen attention. By decorrelating keys in the RKHS, LUCID achieves precise retrieval while operating at standard entropy levels, preserving gradient flow. Theoretical proofs and synthetic experiments confirm that this approach effectively reconciles retrieval precision with optimization stability.

**Contributions**:

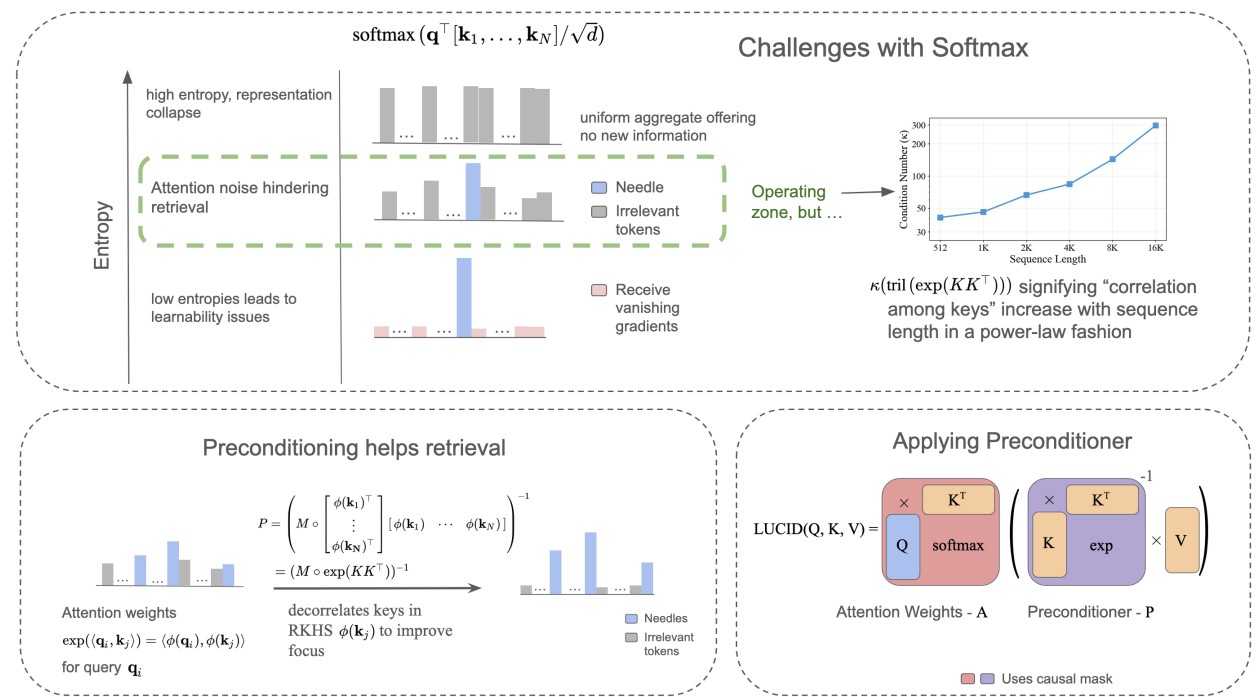

*Figure 1.* **Top:** Challenges with softmax attention. The attention entropy must lie in a narrow operating zone—high entropy leads to uniform aggregation and representation collapse, while low entropy causes vanishing gradients. Even within this zone, correlated keys create attention noise that hinders retrieval of relevant tokens (needles) from irrelevant context. The condition number $\kappa(\text{tril}(\exp(KK^\top)))$ grows with sequence length, indicating increasing key correlation. **Bottom:** LUCID addresses this by constructing a preconditioner $P = (M \circ \exp(KK^\top))^{-1}$, where $\circ$ is element-wise (Hadamard) product and $M$ is the 0-1 causal mask. The preconditioner decorrelates keys in RKHS, sharpening attention on relevant tokens. The resulting attention mechanism (right) combines standard attention weights with the preconditioner using causal masking. The $P^{-1}V$ computation is performed efficiently via `torch.linalg.solve_triangular` (cuBLAS TRSM kernel), exploiting the lower-triangular structure of $P$..

1. **Attention Noise.** We propose a novel preconditioning strategy that decorrelates keys in a Reproducing Kernel Hilbert Space (RKHS), which sharpens attention distributions and reduces noise from irrelevant tokens in long-context scenarios.

2. **Learnability.** We demonstrate that our approach improves the conditioning of the softmax function, addressing critical learnability issues with softmax - vanishing gradient.

3. **Long-Context Results.** Our method achieves superior performance on needle in a haystack benchmarks, outperforming strong baselines such as Path Attention (Yang et al., 2025), DeltaNet (Yang et al., 2024b) and Differential Transformer (Ye et al., 2025).

## 2. LUCID Attention

We derive LUCID Attention by viewing attention mechanisms through the lens of gradient descent in a Reproducing Kernel Hilbert Space (RKHS). This perspective reveals why standard attention accumulates interference and how LU-

CID's preconditioner naturally corrects this limitation. We then describe connections to prior work in the literature.

### 2.1. Deriving LUCID Attention from Gradient Descent in RKHS

#### 2.1.1. PRELIMINARIES AND NOTATION

Let $Q, K, V \in \mathbb{R}^{N \times d}$ denote the query, key, and value matrices, where $N$ is the sequence length and $d$ is the head dimension. Each row $\mathbf{q}_i$, $\mathbf{k}_j$, and $\mathbf{v}_j$ corresponds to the $i$-th or $j$-th query, key, or value vector, respectively. Let the causal mask $M \in \{0, 1\}^{N \times N}$, where $M_{ij} = 1$ if $i \geq j$ and $M_{ij} = 0$ otherwise. We also define $\hat{M} \in \{0, -\infty\}^{N \times N}$, where $\hat{M}_{ij} = 0$ if $i \geq j$ and $\hat{M}_{ij} = -\infty$ otherwise, so that $M = \exp(\hat{M})$.

Following Katharopoulos et al. (2020), the exponential inner product in softmax attention can be expressed using a kernel function:

$$\exp\left(\langle \mathbf{q}_i, \mathbf{k}_j \rangle\right) = \langle \phi(\mathbf{q}_i), \phi(\mathbf{k}_j) \rangle,$$

where $\phi : \mathbb{R}^d \to \mathcal{H}$ is a feature map to a Reproducing Kernel Hilbert Space (RKHS).

### 2.1.2. STANDARD ATTENTION AS GRADIENT DESCENT WITH A LINEAR OBJECTIVE

Consider the causal attention mechanism where at each step $t$, we maintain a state matrix in its linear operator form $S : \mathcal{H} \to \mathbb{R}^d$ that stores key-value associations. We can view the unnormalized attention update (without the softmax denominator) as gradient descent on a *linear objective*:

$$f_t(S) = -\mathbf{v}_t^\top S \phi(\mathbf{k}_t) = -\langle S\phi(\mathbf{k}_t), \mathbf{v}_t \rangle. \qquad (1)$$

The gradient of this objective is $\nabla_S f_t(S) = -\mathbf{v}_t \phi(\mathbf{k}_t)^\top$, and applying gradient descent yields the update rule:

$$S_t = S_{t-1} - \nabla_S f_t(S_{t-1}) = S_{t-1} + \mathbf{v}_t \phi(\mathbf{k}_t)^\top.$$

This is precisely the additive update underlying standard linear attention. After $t$ steps, the retrieval for query $\mathbf{q}_t$ becomes $S_t \phi(\mathbf{q}_t) = \sum_{i=1}^t \mathbf{v}_i \langle \phi(\mathbf{k}_i), \phi(\mathbf{q}_t) \rangle$.

**Limitations of the Linear Objective.** While simple, the linear objective has fundamental drawbacks: **Unbounded from below**: The objective $f_t(S) \to -\infty$ as $\|S\phi(\mathbf{k}_t)\| \to \infty$ in the direction of $\mathbf{v}_t$, providing no natural stopping criterion. **State-independent updates**: The gradient $-\mathbf{v}_t \phi(\mathbf{k}_t)^\top$ is independent of the current state $S_{t-1}$, meaning updates occur regardless of whether the association $\mathbf{k}_t \mapsto \mathbf{v}_t$ is already correctly stored. **Accumulating interference**: Each update adds to the state without removing old information, leading to interference when keys are correlated: $S_t \phi(\mathbf{k}_t) = \mathbf{v}_t + \sum_{i<t} \mathbf{v}_i \langle \phi(\mathbf{k}_i), \phi(\mathbf{k}_t) \rangle$.

### 2.1.3. DERIVING LUCID VIA A QUADRATIC OBJECTIVE

A more principled approach uses a *quadratic objective* that directly measures retrieval error:

$$f_t(S) = \frac{1}{2}\|S\phi(\mathbf{k}_t) - \mathbf{v}_t\|^2. \qquad (2)$$

This objective is bounded below by 0, with a clear minimum at $S\phi(\mathbf{k}_t) = \mathbf{v}_t$. The gradient is:

$$\nabla_S f_t(S) = (S\phi(\mathbf{k}_t) - \mathbf{v}_t)\phi(\mathbf{k}_t)^\top,$$

and applying gradient descent with step size $\beta_t$ gives:

$$S_t = S_{t-1} - \beta_t(S_{t-1}\phi(\mathbf{k}_t) - \mathbf{v}_t)\phi(\mathbf{k}_t)^\top. \qquad (3)$$

**Recurrent Form and Memory Erasure.** Setting $\beta_t = 1$, we can rewrite (3) as:

$$S_t = S_{t-1}(I - \phi(\mathbf{k}_t)\phi(\mathbf{k}_t)^\top) + \mathbf{v}_t\phi(\mathbf{k}_t)^\top. \qquad (4)$$

This is the *delta rule* in the RKHS, which can be interpreted as an erase-then-write mechanism:

$$S_t = S_{t-1} - \underbrace{(S_{t-1}\phi(\mathbf{k}_t))\phi(\mathbf{k}_t)^\top}_{\text{erase old association}} + \underbrace{\mathbf{v}_t\phi(\mathbf{k}_t)^\top}_{\text{write new association}}.$$

This is precisely the update rule underlying DeltaNet (Schlag et al., 2021; Yang et al., 2024b), but generalized from finite-dimensional representation space to the infinite-dimensional RKHS induced by the exponential kernel. In other words, LUCID can be viewed as DeltaNet operating in RKHS.

**Self-Regulation Property.** A key advantage of the quadratic objective is *self-regulation*: when the current prediction is already correct, i.e., $S_{t-1}\phi(\mathbf{k}_t) = \mathbf{v}_t$, the gradient $(S_{t-1}\phi(\mathbf{k}_t) - \mathbf{v}_t)\phi(\mathbf{k}_t)^\top = 0$ and no update occurs. This property is absent in the linear objective, which blindly updates regardless of prediction quality.

**Parallel Form and the Preconditioner.** The recurrent form (4) can be computed in parallel. Collecting outputs for a all the tokens:

$$O = \left(M \circ \phi(Q)\phi(K)^\top\right)\left(I + \text{stril}(\phi(K)\phi(K)^\top)\right)^{-1}V,$$

where $\text{stril}(\cdot)$ denotes the strictly lower triangular part. For the exponential kernel RKHS used in softmax attention, this becomes:

$$O = \left(M \circ \exp(QK^\top)\right)\left(M \circ \exp(KK^\top)\right)^{-1}V.$$

**The LUCID Formulation.** To maintain numerical stability and proper variance scaling in practice, we introduce the standard $1/\sqrt{d}$ logit scaling and RMS normalization for the key vectors inside the preconditioner:

$$\mathbf{k}_{i,\text{RN}} \leftarrow \sqrt{d} \cdot \mathbf{k}_i / \|\mathbf{k}_i\|_2.$$

This normalization ensures that the inner products $\mathbf{k}_{i,\text{RN}}^\top \mathbf{k}_{j,\text{RN}}$ have consistent distributional scale across tokens and the preconditioner matrix is unit-diagonal with controlled off-diagonal magnitudes, yielding better condition numbers. Putting all components together, LUCID Attention is:

$$
\begin{aligned}
\text{LUCID}(Q, K, V) = &\text{softmax}\left(\frac{QK^\top}{\sqrt{d}} + \hat{M}\right) \\
&\cdot \left(M \circ \exp\left(\frac{K_{\text{RN}}K_{\text{RN}}^\top}{\sqrt{d}} - \sqrt{d}\right)\right)^{-1}V.
\end{aligned}
$$
$$(5)$$

**Why LUCID reduces attention noise in longer-contexts?** In the exponential kernel RKHS, a crucial property holds: $\langle \phi(\mathbf{k}_i), \phi(\mathbf{k}_j) \rangle = \exp(\mathbf{k}_i^\top \mathbf{k}_j) > 0$ for all $i, j$. This means keys are *never orthogonal* in feature space, and the preconditioner $(M \circ \exp(KK^\top))^{-1}$ is always non-trivial. The condition number $\kappa$ of this preconditioner quantifies the correction LUCID provides: when $\kappa \approx 1$, keys are approximately orthogonal and linear/quadratic objectives behave

similarly; when $\kappa \gg 1$, keys are highly correlated and LUCID's correction is essential.

We analyze condition numbers on $\sim$1B parameter language models (hidden size 2048, 24 layers, 32 heads) during continual pretraining on the Dolma dataset (Soldaini et al., 2024), fine-tuned from 2K to 65K sequence length. As shown in Figure 2, the condition number grows with sequence length, empirically validating that LUCID's advantage increases for longer sequences where key correlations accumulate.

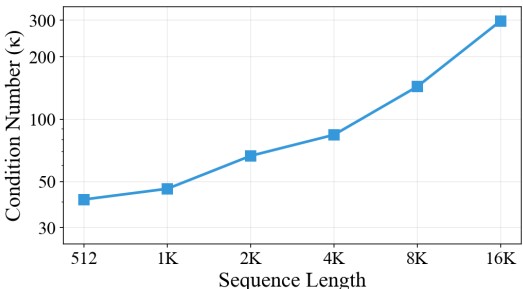

*Figure 2.* Condition number $\kappa$ of the LUCID preconditioner matrix grows with sequence length. Higher $\kappa$ indicates stronger key correlations, where LUCID's correction becomes more essential.

As sequence length increases, the condition number grows due to accumulating key correlations. The cuBLAS TRSM kernel is optimized to handle fp32 outputs, which can accommodate these condition numbers safely. We have successfully trained and evaluated models up to 128K sequence length without stability issues. The RMS normalization of keys ensures the preconditioner matrix has unit diagonal entries and controlled off-diagonal magnitudes, contributing to this stability.

> **From Linear to Quadratic Objectives** LUCID arises from optimizing a quadratic objective (retrieval error) instead of the linear objective underlying standard attention. This yields a self-regulating "erase-then-write" update—the delta rule in RKHS—which is precisely DeltaNet generalized to infinite dimensions.

## 2.2. Learnability of LUCID Attention

A critical requirement for effective attention mechanisms is the ability to represent sharp distributions—when a query $\mathbf{q}_i$ matches a specific key $\mathbf{k}_j$, the attention should concentrate its weight on the corresponding value $\mathbf{v}_j$. However, achieving this sharpness in standard softmax attention creates a fundamental tension with learnability.

**The Softmax Gradient Problem.** Consider a query $\mathbf{q}$ and the softmax attention distribution $\mathbf{a} = \text{softmax}(\mathbf{q}K^\top/\sqrt{d})$. To achieve sharp retrieval, one might decrease the softmax

temperature, leading to:

$$\text{softmax}(\mathbf{z}/\tau) \xrightarrow{\tau \to 0} \mathbf{e}_{\arg\max \mathbf{z}},$$

where $\mathbf{e}_i$ denotes the canonical basis vector with 1 at position $i$ and 0 elsewhere. While this yields a one-hot attention distribution, it introduces a critical problem: the Jacobian of the softmax vanishes. Specifically, the softmax Jacobian is:

$$J = \frac{\partial \mathbf{a}}{\partial \tilde{\mathbf{a}}} = \text{diag}(\mathbf{a}) - \mathbf{a}\mathbf{a}^\top.$$

When $\mathbf{a} = \mathbf{e}_i$ (a one-hot vector), we have:

$$J = \text{diag}(\mathbf{e}_i) - \mathbf{e}_i\mathbf{e}_i^\top = 0.$$

This zero Jacobian means gradients cannot propagate through the attention mechanism, making learning ineffective. This creates a fundamental dilemma: standard attention can either retrieve precisely (low temperature, no gradients) or learn effectively (higher temperature, blurred retrieval).

**LUCID Decouples Retrieval Sharpness from Temperature.** LUCID resolves this tension by achieving sharp retrieval through the preconditioner $(M \circ \exp(KK^\top))^{-1}$ rather than by lowering the softmax temperature. The softmax operates at standard temperature with a well-conditioned gradient, while the preconditioner sharpens the final output through deconvolution. This fundamental decoupling enables LUCID to achieve both precise retrieval and effective learning simultaneously.

**Theorem 1** (Gradient Preservation in LUCID)**.** *Let $\mathbf{o}$ be the LUCID attention output (before multiplying by $V$):*

$$\mathbf{o} = \text{softmax}\left(\frac{\mathbf{q}K^\top}{\sqrt{d}}\right)\left(M \circ \exp\left(\frac{K_{RN}K_{RN}^\top}{\sqrt{d}} - \sqrt{d}\right)\right)^{-1}.$$

*Assume $K \neq 0$ and at least one column of $\text{diag}(\mathbf{a}) - \mathbf{a}\mathbf{a}^\top$ is not in the null-space of $K^\top$, where $\mathbf{a} = \text{softmax}(\mathbf{q}K^\top/\sqrt{d})$. Then $\partial\mathbf{o}/\partial\mathbf{q} \neq 0$.*

*Proof.* The Jacobian of LUCID with respect to $\mathbf{q}$ is:

$$\frac{\partial \mathbf{o}}{\partial \mathbf{q}} = \frac{K^\top}{\sqrt{d}}\left(\text{diag}(\mathbf{a}) - \mathbf{a}\mathbf{a}^\top\right)$$

$$\cdot \left(M \circ \exp\left(\frac{K_{RN}K_{RN}^\top}{\sqrt{d}} - \sqrt{d}\right)\right)^{-1}.$$

Since the preconditioner is a masked lower-triangular matrix with positive diagonal entries, it is invertible with trivial null-space. The gradient vanishes only if $K^\top(\text{diag}(\mathbf{a}) - \mathbf{a}\mathbf{a}^\top) = 0$. This occurs when: (1) $\mathbf{a}$ is one-hot, (2) $K = 0$, or (3) all columns of $\text{diag}(\mathbf{a}) - \mathbf{a}\mathbf{a}^\top$ lie in the null-space of $K^\top$. Since LUCID doesn't operate at extreme temperatures, $\mathbf{a}$ is not one-hot. By assumption, $K \neq 0$ and the column condition holds. Therefore, $\partial\mathbf{o}/\partial\mathbf{q} \neq 0$. $\square$

In summary, LUCID achieves the best of both worlds: precise retrieval similar to zero-temperature softmax (via the preconditioner) while maintaining non-zero gradients (via standard-temperature softmax). This decoupling is the key mechanism that enables LUCID to learn effectively while retrieving precisely.

> **Decoupling Retrieval from Temperature** Standard attention achieves sharp retrieval by lowering temperature, which reduces gradient signal. LUCID decouples these concerns: the preconditioner provides precision while standard temperature preserves gradient flow.

**Synthetic Experiment: Sequential Task Learning.** To empirically validate the learnability advantage of LUCID, we design a two-phase synthetic experiment that isolates the tension between retrieval sharpness and gradient flow. We use a single-layer transformer with model dimension 256 and a single attention head, operating on sequences of length 10 drawn from a vocabulary of 10 digits (0–9).

**Phase 1 (Self-Retrieval):** The model learns to copy the input sequence, i.e., given input $X = (x_1, \ldots, x_{10})$, produce output $Y$ where $y_i = x_i$. This task requires the attention mechanism to learn a sharp, identity-like distribution where each position attends primarily to itself.

**Phase 2 (Cumulative Averaging):** Without resetting weights, the task switches to computing cumulative averages: $y_i = \frac{1}{i} \sum_{j=1}^{i} x_j$. This task requires attending to *all* previous positions with appropriate weights, demanding that the attention mechanism adapt from sparse to dense distributions.

Figure 3 shows the training dynamics. During Phase 1, both standard softmax and LUCID achieve near-zero loss, successfully learning the self-retrieval task. However, the mechanisms by which they achieve this differ fundamentally. The right panel reveals that standard softmax progressively reduces the off-diagonal entries of its Jacobian $(\text{diag}(\mathbf{a}) - \mathbf{a}\mathbf{a}^\top)$ by approximately three orders of magnitude—effectively lowering its implicit temperature to produce sharp attention distributions. LUCID, in contrast, maintains substantially higher Jacobian magnitudes throughout Phase 1, achieving sharpness through its preconditioner rather than temperature reduction.

The consequences become apparent in Phase 2. When the task switches to cumulative averaging, standard softmax struggles to adapt: its near-zero Jacobian prevents gradients from flowing, and the loss remains high. LUCID, having preserved gradient flow during Phase 1, rapidly adapts to the new task, achieving low loss within a few thousand steps. Notably, LUCID's Jacobian magnitude *increases* at the phase transition, demonstrating its ability to dynamically adjust attention patterns when learning requires it.

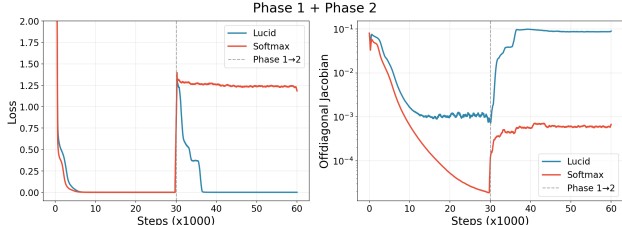

*Figure 3.* **Sequential task learning reveals the learnability-retrieval tradeoff. Left:** Training loss across two phases. Both methods solve Phase 1 (self-retrieval), but only LUCID adapts to Phase 2 (cumulative averaging). **Right:** Off-diagonal Jacobian magnitude (log scale). Standard softmax reduces its Jacobian by $\sim 10^3 \times$ during Phase 1 to achieve sharpness, blocking gradient flow in Phase 2. LUCID maintains higher Jacobian values throughout, enabling rapid adaptation.

### 2.3. LUCID's Efficiency

Implementing LUCID Attention requires solving a triangular linear system involving the masked kernel matrix. Specifically, we solve for $Y \in \mathbb{R}^{N \times d}$ in:

$$\left( M \circ \exp \left( \frac{K_{\text{RN}} K_{\text{RN}}^\top}{\sqrt{d}} - \sqrt{d} \right) \right) Y = V,$$

which can be done efficiently via forward substitution, since the matrix is lower triangular due to the causal mask $M$. Once $Y$ is obtained, the final output is computed by multiplying the softmax attention weights:

$$\text{LUCID}(Q, K, V) = \left( \text{softmax} \left( \frac{QK^\top}{\sqrt{d}} + \hat{M} \right) \right) Y.$$

The overall computational complexity remains $\mathcal{O}(N^2 d)$, similar to standard softmax attention.

**Training Overhead.** We leverage the highly optimized cuBLAS TRSM (Triangular Solve) kernel for solving the triangular system, which enables efficient GPU utilization. Table 1 reports the training time per iteration for LUCID compared to Standard Attention across different sequence lengths. LUCID adds approximately 0-5.5% overhead during training.

*Table 1.* LUCID training overhead across Gemma 3(Team et al., 2025) and Qwen 2.5(Team et al., 2024) remains modest. Modern trends toward grouped-query attention(GQA) with fewer KV heads further reduce LUCID's overhead.

| Architecture | Overhead |
| --- | --- |
| Gemma 3-1B | $\sim 0\%$ |
| Gemma 3-4B | $+3.5\%$ |
| Qwen 2.5-1.5B | $+5.5\%$ |

**Inference Latency.** At inference time, LUCID's overhead becomes negligible. For a 32K context length with batch

size 1 and 100 new tokens generated, LUCID achieves 77ms compared to 76ms for Standard Attention—an overhead of only $\sim$1.3%. This minimal difference arises because the triangular solve operates on the KV cache and can be performed incrementally, as described in the Appendix.

**Compute Ablation.** A natural question is whether the 0-5.5% training overhead of LUCID could be better utilized by simply training the baseline model for longer. To address this, we conducted an ablation study where we increased the training compute for Standard Attention by $\sim$10% additional steps during 64K sequence length continual pretraining, matching LUCID's total compute budget. training the baseline longer does not yield equivalent performance gains—LUCID significantly outperforms the compute-matched baseline in multi-key retrieval by $\sim$20% in Appendix, Table 7.

> **Architectural Gains in Multi-needle, Not Just Compute** LUCID adds 0–5.5% training overhead and $\sim$1.3% inference overhead. Crucially, training the baseline 10% longer does not match LUCID's performance—the gains in multi-needle task stem from architectural design, not extra compute.

## 3. Related Work

**Standard Attention and Long Context Challenges.** The scaled dot-product attention mechanism, introduced by Vaswani et al. (2017), is the cornerstone of the Transformer architecture. Its ability to model long-range dependencies and its parallelizability have driven the success of modern LLMs. The core computation involves Queries - $Q$, Keys - $K$, and Values - $V$, where attention probabilites are derived from $QK^T$ passed through a softmax function.

However, standard attention suffers from $\mathcal{O}(N^2 d)$ computational complexity and performance degradation on long sequences ($N$ large). This degradation manifests as "attention noise", where the softmax forces attention weights onto irrelevant tokens, obscuring the signal from relevant ones (Ye et al., 2025). This is particularly problematic for tasks requiring precise retrieval or reasoning over extended contexts. The Differential Transformer (Diff Transformer) (Ye et al., 2025) attempts to cancels noise by computing the difference between two attention maps ($A_1 - \lambda A_2$), promoting sparsity. In contrast, LUCID attention tries to solve the root cause of the attention noise issue - correlated keys. In LUCID, a preconditioner developed from key-key similarities is used, which decorrelates the keys and removes attentional noise. This preconditioning sharpens the attention distribution and enable precise retrieval.

*Table 2.* Comparing mathematical formula of different attention mechanisms. Here, $Q, K, V \in \mathbb{R}^{N \times d}$ are queries, keys, and values, $\lambda \in \mathbb{R}, \boldsymbol{\beta} \in \mathbb{R}^N, W \in \mathbb{R}^{N \times d}$ are learnable parameters. $M$ and $\hat{M}$ are multiplicative and additive causal masks, respectively. $T = \text{diag}(\boldsymbol{\beta})^{-1} + \text{stril}(WW^\top)$. Let diag, tril, and stril be diagonal, lower-triangular and strictly-lower-triangular retention operators, respectively.

| Model | Attention Formula |
|---|---|
| **Standard** | $\text{softmax}\left(QK^\top + \hat{M}\right)V$ |
| **Diff Trans.** | $\left(\text{softmax}\left(Q_1 K_1^\top + \hat{M}\right) - \lambda\,\text{softmax}\left(Q_2 K_2^\top + \hat{M}\right)\right)V$ |
| **DeltaNet** | $\left(M \circ QK^\top\right)\left(I + \text{diag}(\boldsymbol{\beta})\text{stril}(KK^\top)\right)^{-1}\text{diag}(\boldsymbol{\beta})V$ |
| **PaTH** | $\text{softmax}\left(\text{tril}(QK^\top) - \text{tril}(QW^\top)T^{-1}\text{stril}(WK^\top)\right)V$ |
| **LUCID** | $\text{softmax}\left(QK^\top + \hat{M}\right)\left(M \circ \exp\left(KK^\top\right)\right)^{-1}V$ |

**Linear Complexity Alternatives.** Another line of research seeks to overcome the $\mathcal{O}(N^2)$ bottleneck by developing linear complexity ($\mathcal{O}(N)$) attention mechanisms. These often replace softmax with simpler kernels and reformulate the computation as a recurrent process, eliminating the need for a large KV cache (Katharopoulos et al., 2020; Yang et al., 2024b). While efficient, early linear attention variants often underperformed softmax attention, particularly in recall. DeltaNet (Yang et al., 2024b) improves recall within this linear framework by replacing the simple additive memory update with one inspired by the delta rule, allowing for more targeted memory modification. GatedDeltaNet (Yang et al., 2024a) further enhances this by combining the delta rule with a data-dependent gating mechanism (similar to Mamba2 (Dao & Gu, 2024)) for adaptive memory erasure and targeted updates. LUCID Attention is distinct from these approaches as it retains the $\mathcal{O}(N^2 d)$ complexity, focusing on improving the quality of standard attention mechanism rather than achieving efficiency. PaTH Attention (Yang et al., 2025) introduces a data-dependent positional encoding mechanism via preconditioners, developed as a generatlization of the DeltaNet.

## 4. Experimental Setup

We compare LUCID Attention against different baselines: Standard Attention, Diff Transformer, DeltaNet, and PaTH Attention. We train, and fine-tune our models on a subset of Dolma dataset ($\sim$6.5B tokens) provided by Allen AI (Soldaini et al., 2024) and use it for Needle-In-A-Haystack (NIAH) evaluations and downstream tasks.

### 4.1. Model Configuration

For NIAH evaluations and downstream tasks, we train decoder-only transformers with 22 layers, model dimension 2048, MLP dimension 5632, and vocabulary size 32,000. Each model uses 32 attention heads and 4 key-value heads

*Table 3.* Performance comparison of different models against LUCID Attention for MNIAH task at 2048 sequence length.

| Model | Number of Needles | | |
|---|---|---|---|
| | 2 | 4 | 6 |
| Standard | 74.2 | 51.0 | 38.8 |
| Diff Trans. | 72.4 | 35.4 | 26.0 |
| DeltaNet | 37.0 | 16.8 | 11.0 |
| Path | 61.4 | 44.2 | 37.2 |
| **LUCID** | **76.6** | **55.8** | **43.6** |

with head dimension of 68. PaTH Rotary positional embeddings - RoPE (Su et al., 2024) are used in all attention variants except for DeltaNet and PaTH. RoPE frequencey is set to 10,000 during pretraining and was changed to 64,000 during fine-tuning. We use the AdamW optimizer (Kingma & Ba, 2014) with learning rate $5 \times 10^{-4}$, $\beta_1 = 0.9$, and $\beta_2 = 0.99$ with weight decay of 0.01. We conduct pretraining of all models on sequence length 2048 for 11k optimization steps with each step having effective batch size of 256 sequences. Then we fine-tune on sequence length 65536 for 500 optimization steps with each step having effective batch size of 16. We use Huggingface Transformers library (Wolf et al., 2020) to train our models.

## 4.2. Needle-In-A-Haystack Evaluations

We conduct experiments on SNIAH and MNIAH tasks before and after fine-tuning on varying sequence lengths and number of needles. For these tasks, we modified the RULER tasks (Hsieh et al., 2024) in the eval-harness framework (Gao et al., 2024). Note that RULER has multiple SNIAH and MNIAH tasks. Here, we are reporting average accuracies. Throughout the NIAH evaluations, LUCID Attention shows better retrieval capability compared to Standard Attention both for varying sequence lengths and varying number of needles. We also vary the finetuning sequence length and observe that LUCID scales better with finetuning sequence length compared to standard attention.

## 4.3. Long-Context Reasoning: BABILong

We evaluate on BABILong (Kuratov et al., 2024), a benchmark designed to test language models on "needle-in-a-haystack" reasoning tasks at scale. BABILong extends the classic bAbI tasks (Weston et al., 2015) by embedding reasoning problems within long distractor texts sampled from PG19. We focus on QA1–QA5, which test fact retrieval and multi-hop reasoning. We finetune all models on the BABILong training set and evaluate at three context lengths: 32K, 64K, and 128K. We introduce a variant of LUCID called LUCID-PATH, which combines LUCID's key decorrelation mechanism with PaTH positional encoding (Yang et al., 2025), enabling length extrapolation to contexts longer than

those seen during training. The integration applies LUCID's preconditioner to the PaTH attention scores via a block-wise forward-substitution algorithm (see Appendix A.3 for details).

Figure 6 presents the average accuracy across QA1–QA5. The results reveal a striking difference in how methods scale with context length. Standard and Diff attention exhibit rapid performance degradation, dropping from ~0.14 at 32K to near zero at 128K. Path attention shows moderate degradation but maintains some capability at longer contexts. In contrast, Lucid-Path demonstrates remarkable stability: accuracy remains between 0.21–0.25 across all context lengths, with minimal degradation from 32K to 128K. Notably, Lucid-Path achieves this with consistently low variance across tasks (tight shaded regions), indicating robust performance on both single-hop (QA1) and multi-hop (QA2–QA3) retrieval. These results suggest that Lucid-Path's attention mechanism more effectively preserves access to distributed facts in long sequences, a critical capability for real-world long-document understanding.

## 4.4. Long-Context Understanding: LongBench and SCROLLS

To evaluate performance on realistic long-context tasks, we benchmark on LongBench (Bai et al., 2023) and SCROLLS (Shaham et al., 2022), covering multi-document QA, single-document QA, and summarization. We evaluate on six tasks spanning three categories: multi-document QA (2WikiMQA, HotpotQA), single-document QA (Multi-fieldQA, Qasper), and summarization (QMSum). We compare against Standard softmax attention, Differential attention (Ye et al., 2025), PaTH attention, and linear attention variants including DeltaNet (Yang et al., 2024b), GLA (Yang et al., 2024a), and GSA (Zhang et al., 2024). Detailed task descriptions are provided in Section A.6.

Table 4 presents the results. LUCID-PaTH achieves the best performance on four of six tasks, with particularly strong gains on Qasper (+1.14 F1 over PaTH) and QMSum ROUGE-1 (+0.22 over PaTH). LUCID alone already outperforms most baselines, achieving the best HotpotQA F1 (0.0862) and QMSum ROUGE-L (12.60). Linear attention variants (DeltaNet, GLA, GSA) underperform substantially, particularly on multi-document QA where cross-document reasoning is critical—DeltaNet achieves only 0.036 F1 on 2WikiMQA compared to 0.274 for LUCID.

## 4.5. Empirical Analysis of Attention Noise Reduction

To provide direct empirical evidence that LUCID reduces attention noise, we analyze the *hitrate*—the fraction of attention weight assigned to semantically relevant tokens during retrieval tasks. Specifically, on NIAH tasks, we measure the average attention weight placed on the "needle" tokens that

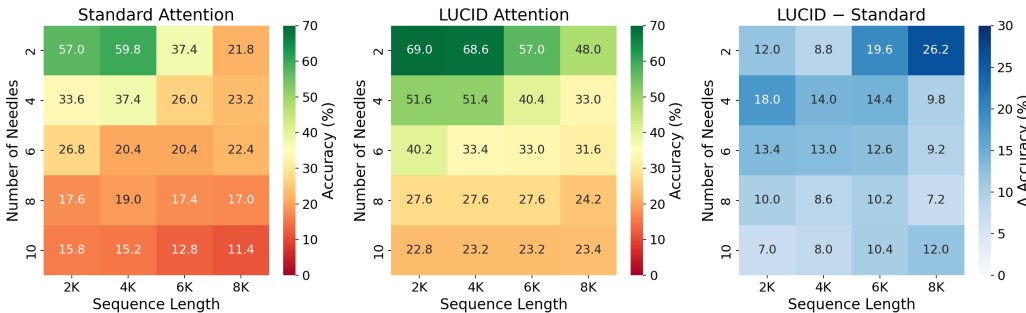

*Figure 4.* Performance comparison on MNIAH across varying number of needles and sequence lengths. **Left:** Standard Attention accuracy degrades sharply as task difficulty increases (more needles, longer sequences), dropping to 11.4% in the hardest configuration. **Middle:** LUCID Attention maintains substantially higher accuracy across all settings. **Right:** The difference highlights consistent improvements of 10–26%, with LUCID providing the largest gains at longer sequence lengths where Standard Attention struggles most.

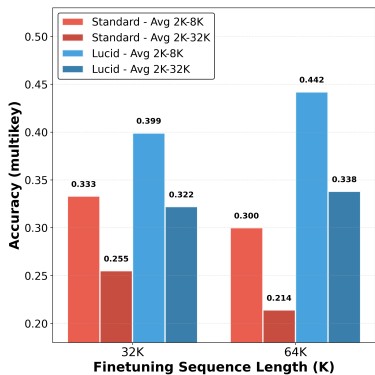

*Figure 5.* **Multi-needle retrieval accuracy improves with longer finetuning for LUCID.** Models finetuned at 32K and 64K sequence lengths are evaluated on multi-needle tasks with contexts averaged over 2K-8K and 2K-32K. The performance gap between LUCID and Standard Attention increases from +19.8% (32K finetuning) to +47.3% (64K finetuning).

contain the answer.

LUCID Attention achieves a hitrate of 0.2845 compared to 0.1817 for Standard Attention—a relative improvement of 56.6%. This substantial increase demonstrates that LUCID's preconditioning effectively reduces attention noise by concentrating probability mass on the relevant tokens rather than dispersing it across irrelevant context. This empirical result corroborates our theoretical motivation: by decorrelating keys in the RKHS feature space, LUCID enables queries to focus more precisely on important keys.

## 5. Conclusion

Standard softmax attention faces fundamental challenges in long-context settings: correlated keys lead to diffuse attention distributions, and lowering temperature to sharpen focus causes vanishing gradients. LUCID Attention addresses these issues by preconditioning with a key-key similarity matrix, decorrelating keys in RKHS to improve attention

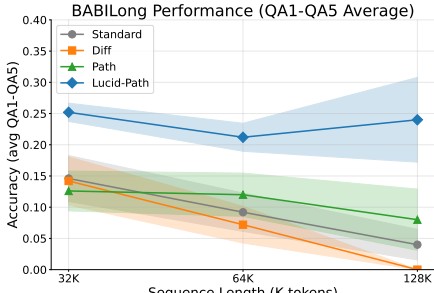

*Figure 6.* **BABILong long-context retrieval performance.** Tests multi-hop fact retrieval across increasingly long contexts (32K–128K tokens). While baseline methods exhibit substantial performance degradation as sequence length increases, Lucid-Path maintains stable accuracy.

*Table 4.* **LongBench and SCROLLS evaluation results.** F1 scores for QA tasks and ROUGE scores for summarization. Best results are in **bold**, second-best are underlined.

| Model | Multi-Doc QA | | Single-Doc QA | | Summ. (32K) | |
|---|---|---|---|---|---|---|
| | 2WikiMQA | HotpotQA | MultifieldQA | Qasper | R-1 | R-L |
| Standard | 0.240 | 0.073 | 0.113 | 7.69 | 11.79 | 10.39 |
| Diff | 0.263 | 0.081 | 0.139 | 8.70 | 13.19 | 11.07 |
| DeltaNet | 0.036 | 0.019 | 0.076 | 7.06 | 10.74 | 8.72 |
| GLA | 0.228 | 0.045 | 0.114 | 8.17 | 10.97 | 9.20 |
| GSA | 0.232 | 0.051 | 0.116 | 7.54 | 9.11 | 7.81 |
| PaTH | **0.283** | 0.079 | 0.139 | 10.57 | 14.62 | 12.00 |
| LUCID | 0.274 | **0.086** | 0.144 | 10.55 | 14.79 | **12.60** |
| LUCID-PaTH | 0.275 | 0.085 | **0.149** | **11.70** | **14.83** | 12.57 |

focus while preserving full $\mathcal{O}(N^2 d)$ complexity. Our experiments demonstrate substantial gains on long-context retrieval tasks—up to 18% on BABILong and 14% on RULER multi-needle—with minimal overhead. **Limitations.** Our work focuses on causal language modeling, where the triangular structure of the preconditioner matrix enables efficient linear solves with minimal overhead. However, for bidirectional settings such as diffusion models, the preconditioner loses its triangular structure, making the linear solve computationally expensive—exploring efficient solutions for this case remains an important direction for future work.

## Acknowledgements

We thank Manzil Zaheer, Devvrit Khatri, Rohan Anil, and Vineet Gupta for helpful discussions.

## Impact Statement

This paper presents work whose goal is to advance the field of Machine Learning. There are many potential societal consequences of our work, none which we feel must be specifically highlighted here.

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

# A. Appendix

**Table of Contents**

### A.1. KV Caching

To enable LUCID for auto-regressive behavior, we need to cache unnormalized keys $K_{\text{past}} \in \mathbb{R}^{L_{\text{past}} \times d}$ and the solution of the triangular solver $(M \circ \exp(K_{\text{RN(past)}} K_{\text{RN(past)}}^\top / \sqrt{d} - \sqrt{d})) Y_{\text{past}} = V_{\text{past}}$. When new keys $K_{\text{new}} \in \mathbb{R}^{L_{\text{new}} \times d}$ and values $V_{\text{new}} \in \mathbb{R}^{L_{\text{new}} \times d}$ are generated during decoding, new solutions of the triangular system $Y_{\text{new}} \in \mathbb{R}^{L_{\text{new}} \times d}$ can be easily computed by solving smaller triangular system

$$\left( M \circ \exp\left( \frac{K_{\text{RN(new)}} K_{\text{RN(new)}}^\top}{\sqrt{d}} - \sqrt{d} \right) \right) Y_{\text{new}} = V_{\text{new}} - \exp\left( \frac{K_{\text{RN(new)}} K_{\text{RN(past)}}^\top}{\sqrt{d}} - \sqrt{d} \right) Y_{\text{past}}.$$

Once $Y_{\text{new}}$ is obtained, the new outputs of LUCID Attention can be computed through the standard softmax KV caching mechanism. Asymptotically, the auto-regressive complexity of LUCID Attention is same as that of Standard Attention.

**Decoding Details.** During auto-regressive decoding, we generate one token at a time ($L_{\text{new}} = 1$). The KV cache stores the unnormalized keys $K_{\text{past}}$ and the preconditioned values $Y_{\text{past}}$ from all previous tokens. For each new token, we receive the query $Q_{\text{new}} \in \mathbb{R}^{1 \times d}$, key $K_{\text{new}} \in \mathbb{R}^{1 \times d}$, and value $V_{\text{new}} \in \mathbb{R}^{1 \times d}$.

The decoding process proceeds in two stages. First, we compute the preconditioned value $Y_{\text{new}}$ for the new token. We apply RMSNorm to obtain the normalized keys $K_{\text{RN(new)}} = \text{RMSNorm}(K_{\text{new}})$ and $K_{\text{RN(past)}} = \text{RMSNorm}(K_{\text{past}})$. Note that $K_{\text{RN(past)}}$ can be cached to avoid recomputation. For single-token decoding, the diagonal block of the preconditioner matrix is simply $\exp(0) = 1$ (since $K_{\text{RN(new)}} K_{\text{RN(new)}}^\top / \sqrt{d} - \sqrt{d} = 1 - 1 = 0$ when RMS normalized), so no triangular solve is needed. The preconditioned value simplifies to:

$$Y_{\text{new}} = V_{\text{new}} - \exp\left( \frac{K_{\text{RN(new)}} K_{\text{RN(past)}}^\top}{\sqrt{d}} - \sqrt{d} \right) Y_{\text{past}}.$$

Second, we compute the attention output $O_{\text{new}}$ using the standard softmax attention mechanism with the cached keys and the preconditioned values:

$$O_{\text{new}} = \text{softmax}\left( \frac{Q_{\text{new}} [K_{\text{past}}; K_{\text{new}}]^\top}{\sqrt{d}} \right) [Y_{\text{past}}; Y_{\text{new}}].$$

Finally, we update the KV cache by appending $K_{\text{new}}$ and $Y_{\text{new}}$ to the cached keys and preconditioned values, respectively. The complete decoding procedure is summarized in Algorithm 1.

The key insight is that LUCID's preconditioner can be applied incrementally during decoding. The computational overhead compared to standard attention is a single matrix-vector multiplication to compute $\exp(S) \cdot Y_{\text{past}}$, which is $O(L_{\text{past}} \cdot d)$ the same complexity as the standard attention computation over the cached keys. This makes LUCID's inference overhead negligible in practice.

---

**Algorithm 1** LUCID Attention Decoding

---

1: **Input:** $Q_{\text{new}} \in \mathbb{R}^{B \times H_Q \times 1 \times D}, K_{\text{new}} \in \mathbb{R}^{B \times H \times 1 \times D}, V_{\text{new}} \in \mathbb{R}^{B \times H \times 1 \times D}, \sqrt{d}, \text{KV\_Cache}$
2: **Output:** $O_{\text{new}} \in \mathbb{R}^{B \times H_Q \times 1 \times D}$, updated KV_Cache
3:
4: // Retrieve cached keys and preconditioned values
5: $(K_{\text{past}}, Y_{\text{past}}) \leftarrow \text{KV\_Cache}, \qquad K_{\text{past}}, Y_{\text{past}} \in \mathbb{R}^{B \times H \times L_{\text{past}} \times D}$
6:
7: // Stage 1: Compute preconditioned value $Y_{\text{new}}$
8: $K_{\text{RN(new)}} \leftarrow \text{RMSNorm}(K_{\text{new}})$
9: $K_{\text{RN(past)}} \leftarrow \text{RMSNorm}(K_{\text{past}})$
10: $S \leftarrow K_{\text{RN(new)}} \cdot K_{\text{RN(past)}}^{\top}/\sqrt{d} - \sqrt{d}$
11: $Y_{\text{new}} \leftarrow V_{\text{new}} - \exp(S) \cdot Y_{\text{past}}$
12:
13: // Stage 2: Compute attention output using standard decoding
14: $K_{\text{full}} \leftarrow [K_{\text{past}}; K_{\text{new}}]$
15: $Y_{\text{full}} \leftarrow [Y_{\text{past}}; Y_{\text{new}}]$
16: $O_{\text{new}} \leftarrow \text{Attention\_Decoding}(Q_{\text{new}}, K_{\text{full}}, Y_{\text{full}})$
17:
18: // Update KV cache with new key and preconditioned value
19: $\text{KV\_Cache} \leftarrow (K_{\text{full}}, Y_{\text{full}})$
20:
21: **return** $O_{\text{new}}$, KV_Cache

---

## A.2. Memory Efficient Implementation of LUCID

We develop a memory friendly block-wise algorithms for both the forward and backward passes, exploiting the Gram matrix structure of the LUCID preconditioner matrix $P = \exp(KK^{\top}/\sqrt{d} - \sqrt{d})$. This enables processing of arbitrarily long sequences within GPU memory constraints. The forward pass of the block-wise implementation is presented in Algorithm 2.

---

**Algorithm 2** Block-wise Forward Pass: Lower Triangular System Solver

---

1: **Input:** $K_{RN} \in \mathbb{R}^{B \times H \times L \times D}, V \in \mathbb{R}^{B \times H \times L \times D}, \sqrt{d}, BS$
2: **Output:** $Y \in \mathbb{R}^{B \times H \times L \times D}$
3: $L \leftarrow \text{length}(K_{RN})$
4: $Y \leftarrow \text{zeros\_like}(V)$
5: **for** $i = 0$ **to** $L - BS$ **by** $BS$ **do**
6: $\quad$ rhs $\leftarrow V[i : i + BS]$
7: $\quad$ // Sequential inner loop from start to current block
8: $\quad$ **for** $j = 0$ **to** $i - BS$ **by** $BS$ **do**
9: $\qquad \exp_{ij} \leftarrow \exp(K_{RN}[i : i + BS] \cdot K_{RN}[j : j + BS]^T/\sqrt{d} - \sqrt{d})$
10: $\qquad$ rhs $\leftarrow$ rhs $- \exp_{ij} \cdot Y[j : j + BS]$
11: $\quad$ **end for**
12: $\quad$ // Solve diagonal block
13: $\quad \exp_{ii} \leftarrow \exp(K_{RN}[i : i + BS] \cdot K_{RN}[i : i + BS]^T/\sqrt{d} - \sqrt{d})$
14: $\quad Y[i : i + BS] \leftarrow \texttt{torch.linalg.solve\_triangular}(\exp_{ii}, \text{rhs}, \text{lower=true}, \text{unitdiagonal=true})$
15: **end for**
16: **return** $Y$

---

**Block-wise Forward Pass.** Algorithm 2 solves the lower-triangular system $(I + L)Y = V$ in a block-wise manner, where $L$ is strictly lower-triangular with entries $L_{ij} = \exp(K_{RN(i)}K_{RN(j)}^{\top}/\sqrt{d} - \sqrt{d})$ for $i > j$. The algorithm processes blocks sequentially from top to bottom (line 5), maintaining the causal dependency structure of the forward-substitution process.

For each row block $i$, we initialize the right-hand side with the corresponding block of values $V[i : i + BS]$ (line 6). The inner loop (lines 8-11) iterates through all previous column blocks $j < i$, computing the block attention score $\exp_{ij}$ and

subtracting the contribution $\exp_{ij} \cdot Y[j : j + BS]$ from the right-hand side. This removes the influence of all earlier blocks, isolating the contribution that needs to be solved for. After processing all previous blocks, we solve the diagonal block system using the triangular solver (lines 13-14) to obtain $Y[i : i + BS]$. The diagonal block uses the unit diagonal option since the implicit diagonal of $(I + L_{ii})$ is identity. This block-wise forward-substitution is mathematically equivalent to the full solve but operates on manageable block sizes.

For the backward pass, following the derivations in (Giles, 2008), we obtain the gradients $\nabla_{K_{RN}}$ and $\nabla_V$.

$$\nabla_V = (P^\top)^{-1} \nabla_Y$$

$$\nabla_{K_{RN}} = -\frac{1}{\sqrt{d}} \left( \nabla_V Y^\top \odot P + Y \nabla_V^\top \odot P \right) K_{RN}$$

The gradient with respect to $V$ involves solving a transposed triangular system, similar to the forward pass but processing blocks in reverse order. The gradient with respect to $K_{RN}$ is more involved, as it requires computing contributions from both $\nabla_V Y^\top \odot P$ and $Y \nabla_V^\top \odot P$, where $P = (I + L)^{-1}$ is the lower-triangular preconditioner. The Hadamard products with $P$ must be carefully computed in a block-wise manner to avoid materializing the full matrix. The block-wise implementations for computing $\nabla_V$ and $\nabla_{K_{RN}}$ are provided in Algorithm 3 and Algorithm 4, respectively.

---

**Algorithm 3** Block-wise Backward Pass: Gradient Computation for Values

1: **Input:** $K_{RN} \in \mathbb{R}^{B \times H \times L \times D}, \nabla_Y \in \mathbb{R}^{B \times H \times L \times D}, \sqrt{d}, BS$
2: **Output:** $\nabla_V \in \mathbb{R}^{B \times H \times L \times D}$
3: $L \leftarrow \text{length}(K_{RN})$
4: $\nabla_V \leftarrow \text{zeros\_like}(K_{RN})$
5: **for** $i = L - BS$ **downto** 0 **by** $BS$ **do**
6:     $\text{rhs} \leftarrow \nabla_Y[i : i + BS]$
7:     // Sequential inner loop from end to current block
8:     **for** $j = L - BS$ **downto** $i + BS$ **by** $BS$ **do**
9:       $\exp_{ji} \leftarrow \exp(K_{RN}[j : j + BS] \cdot K_{RN}[i : i + BS]^T / \sqrt{d} - \sqrt{d})$
10:      $\text{rhs} \leftarrow \text{rhs} - \exp_{ji}^T \cdot \nabla_V[j : j + BS]$
11:     **end for**
12:     // Solve diagonal block
13:     $\exp_{ii} \leftarrow \exp(K_{RN}[i : i + BS] \cdot K_{RN}[i : i + BS]^T / \sqrt{d} - \sqrt{d})$
14:     $\nabla_V[i : i + BS] \leftarrow \texttt{torch.linalg.solve\_triangular}(\exp_{ii}, \text{rhs}, \text{lower=false}, \text{unitdiagonal=true})$
15: **end for**
16: **return** $\nabla_V$

---

**Computing $\nabla_V$ via Backward-Substitution.** Algorithm 3 solves the transposed triangular system $(I + L)^\top \nabla_V = \nabla_Y$ to compute the gradient with respect to the input values. Since $(I + L)$ is lower-triangular, its transpose is upper-triangular, requiring backward-substitution that processes blocks from bottom to top (line 5).

For each row block $i$, we initialize the right-hand side with $\nabla_Y[i : i + BS]$ (line 5). The inner loop (lines 8-11) iterates through all later blocks $j > i$ in reverse order, computing $\exp_{ji}^\top$ and subtracting its contribution from the right-hand side. This propagates gradients from later blocks back to the current block. After accumulating contributions from all later blocks, we solve the diagonal block system using the transposed triangular solver (lines 14), with `lower=false` to indicate an upper-triangular solve. This backward-substitution mirrors the structure of the forward pass but processes blocks in reverse order.

**Computing $\nabla_{K_{RN}}$ with Mirrored Block Processing.** Algorithm 4 computes the gradient with respect to the key matrix $K_{RN}$ using the formula $\nabla_{K_{RN}} = -\frac{1}{\sqrt{d}} \left( \nabla_V Y^\top \odot P + Y \nabla_V^\top \odot P \right) K_{RN}$. The key challenge is efficiently computing the two terms $\nabla_V Y^\top \odot P$ and $Y \nabla_V^\top \odot P$ in a block-wise manner. The algorithm employs a mirroring strategy to process both the lower-triangular part (from $\nabla_V Y^\top \odot P$) and the upper-triangular part (from $Y \nabla_V^\top \odot P$) simultaneously, reducing the number of block loads.

The outer loop (line 5) processes blocks from the beginning of the sequence. For each block $i$, the algorithm computes

---

**Algorithm 4** Block-wise Backward Pass: Gradient Computation for Keys

---

1: **Input:** $K_{RN} \in \mathbb{R}^{B \times H \times L \times D}, Y \in \mathbb{R}^{B \times H \times L \times D}, \nabla_V \in \mathbb{R}^{B \times H \times L \times D}, \sqrt{d}, BS$
2: **Output:** $\nabla_{K_{RN}} \in \mathbb{R}^{B \times H \times L \times D}$
3: $L \leftarrow \text{length}(K_{RN})$
4: $\nabla_{K_{RN}} \leftarrow \text{zeros\_like}(K_{RN})$
5: **for** $i = 0$ **to** $L - BS$ **by** $BS$ **do**
6:    $\exp_{ii} \leftarrow \exp(K_{RN}[i : i + BS] \cdot K_{RN}[i : i + BS]^T / \sqrt{d} - \sqrt{d})$
7:    $M_{ii} \leftarrow \nabla_V[i : i + BS] \cdot Y[i : i + BS]^T$
8:    $k_1 \leftarrow (M_{ii} \odot \text{tril}(\exp_{ii})) \cdot K_{RN}[i : i + BS]$
9:    // Compute corresponding block from end
10:    $i_{end} \leftarrow L - (i + BS)$
11:    $\exp_{i_{end} i_{end}} \leftarrow \exp(K_{RN}[i_{end} : i_{end} + BS] \cdot K_{RN}[i_{end} : i_{end} + BS]^T / \sqrt{d} - \sqrt{d})$
12:    $M_{i_{end} i_{end}} \leftarrow Y[i_{end} : i_{end} + BS] \cdot \nabla_V[i_{end} : i_{end} + BS]^T$
13:    $k_2 \leftarrow (M_{i_{end} i_{end}} \odot \text{triu}(\exp_{i_{end} i_{end}})) \cdot K_{RN}[i_{end} : i_{end} + BS]$
14:    // Accumulate off-diagonal contributions
15:    **for** $j = 0$ **to** $i - BS$ **by** $BS$ **do**
16:      $\exp_{ij} \leftarrow \exp(K_{RN}[i : i + BS] \cdot K_{RN}[j : j + BS]^T / \sqrt{d} - \sqrt{d})$
17:      $M_{ij} \leftarrow \nabla_V[i : i + BS] \cdot Y[j : j + BS]^T$
18:      $k_1 \leftarrow k_1 + (M_{ij} \odot \exp_{ij}) \cdot K_{RN}[j : j + BS]$
19:      $j_{end} \leftarrow L - (j + BS)$
20:      $\exp_{j_{end} i_{end}} \leftarrow \exp(K_{RN}[j_{end} : j_{end} + BS] \cdot K_{RN}[i_{end} : i_{end} + BS]^T / \sqrt{d} - \sqrt{d})$
21:      $M_{i_{end} j_{end}} \leftarrow Y[i_{end} : i_{end} + BS] \cdot \nabla_V[j_{end} : j_{end} + BS]^T$
22:      $k_2 \leftarrow k_2 + (M_{i_{end} j_{end}} \odot \exp_{j_{end} i_{end}}^T) \cdot K_{RN}[j_{end} : j_{end} + BS]$
23:    **end for**
24:    $\nabla_{K_{RN}}[i : i + BS] \leftarrow \nabla_{K_{RN}}[i : i + BS] + k_1$
25:    $\nabla_{K_{RN}}[i_{end} : i_{end} + BS] \leftarrow \nabla_{K_{RN}}[i_{end} : i_{end} + BS] + k_2$
26: **end for**
27: $\nabla_{K_{RN}} \leftarrow -\nabla_{K_{RN}} \cdot \sqrt{d}$
28: **return** $\nabla_{K_{RN}}$

---

contributions from both the lower-triangular term at block $i$ (lines 6-8) and the corresponding upper-triangular term at the mirrored position $i_{end} = L - (i + BS)$ from the end (lines 10-13). The diagonal blocks require special handling with `tril` and `triu` masks (lines 8 and 13) to separate the lower and upper triangular contributions.

The inner loop (lines 15-23) accumulates off-diagonal contributions for both positions simultaneously. For each earlier block $j < i$, we compute the lower-triangular contribution $M_{ij} \odot \exp_{ij}$ for position $i$ (lines 16-18) and the upper-triangular contribution $M_{i_{end} j_{end}} \odot \exp_{j_{end} i_{end}}^\top$ for the mirrored position (lines 19-22). This mirroring strategy exploits symmetry in the computation, processing two blocks per iteration and reducing memory traffic. Finally, the accumulated gradients are scaled by $-1/\sqrt{d}$ (line 27) to account for the normalization factor in the gradient formula.

### A.3. LUCID-PaTH

To enable length extrapolation in BABILong tasks, we incorporated PaTH positional encoding (Yang et al., 2025) in LUCID by modifying the PaTH-attention triton kernel. The key challenge in integrating PaTH with LUCID lies in efficiently computing the preconditioned values while respecting the causal structure imposed by PaTH's attention mechanism.

The LUCID-PaTH preconditioner employs a block-wise forward-substitution algorithm, similar to solving a lower-triangular linear system. We denote the block size as $BS$, which divides the sequence length $L$ into $N_{blocks} = \lceil L/BS \rceil$ blocks. The core idea is to iteratively refine each block of the output $Y$ by removing the influence of all previous blocks, weighted by the exponential of their PaTH attention scores. This process ensures that the preconditioned values $Y$ properly account for the structured dependencies encoded in the PaTH positional embeddings. The pseudo-code for this preconditioner is shown in Algorithm 5.

The algorithm operates as follows. We initialize $Y$ as a copy of the value matrix $V$ (line 3), which serves as our starting

---

**Algorithm 5** LUCID-PaTH preconditioner using Block-wise Forward-Substitution.

---

1: **Input:** $K_{RN} \in \mathbb{R}^{B \times H \times L \times D}, V \in \mathbb{R}^{B \times H \times L \times D}, W \in \mathbb{R}^{B \times H \times L \times D}, \boldsymbol{\beta} \in \mathbb{R}^{B \times H \times L}, \sqrt{d}$
2: **Output:** $Y \in \mathbb{R}^{B \times H \times L \times D}$
3: $Y \leftarrow V.\text{clone}()$
4: $N_{blocks} \leftarrow \text{num\_row\_blocks}$
5: **for** $i = 0$ **to** $N_{blocks} - 1$ **do**
6:     $S_{ii} \leftarrow \text{PaTH\_Score}(K_{RN}, K_{RN}, W, \boldsymbol{\beta}, i, i)/\sqrt{d} - \sqrt{d}$
7:     // Sequential inner loop from diagonal to the left
8:     **for** $j = i - 1$ **downto** $0$ **do**
9:         $S_{ij} \leftarrow \text{PaTH\_Score}(K_{RN}, K_{RN}, W, \boldsymbol{\beta}, i, j)/\sqrt{d} - \sqrt{d}$
10:        $Y[i] \leftarrow Y[i] - \exp(S_{ij})Y[j]$
11:     **end for**
12:     // Normalize current row block with diagonal solver
13:     $O[i] \leftarrow \texttt{torch.linalg.solve\_triangular}(\exp(S_{ii}), O[i], \text{lower=true}, \text{unitdiagonal=true})$
14: **end for**
15: **return** $Y$

---

point for the forward-substitution process. The outer loop (lines 5-14) iterates over each row block $i$ from top to bottom, processing them sequentially. For each row block $i$, we first compute the diagonal block's PaTH attention score $S_{ii}$ (line 6), which captures the self-attention within the block.

The inner loop (lines 8-11) implements the forward-substitution step, iterating backwards through all previous column blocks $j < i$. For each pair $(i, j)$, we compute the PaTH attention score $S_{ij}$ and subtract the contribution of block $j$ from block $i$ of the output: $Y[i] \leftarrow Y[i] - \exp(S_{ij})Y[j]$. This subtraction removes the influence of earlier blocks, effectively isolating the contribution that needs to be normalized. The backward iteration order is crucial: following (Yang et al., 2025), for any row block, column blocks must be generated from the diagonal block to the left-most block to maintain the proper causal dependencies in PaTH attention.

After processing all previous blocks, we normalize the current row block $Y[i]$ using the diagonal block scores (line 13). Specifically, we solve the lower-triangular system $\exp(S_{ii}) \cdot Y[i] = O[i]$ using triangular solver with unit diagonal. This diagonal normalization completes the preconditioning for block $i$, ensuring that the block is properly scaled according to its self-attention pattern.

Here, the $\text{PaTH\_Score}(Q, K, W, \boldsymbol{\beta}, i, j)$ routine generates the $[i, j]$ block of the PaTH-attention score matrix, defined as:

$$\text{PaTH\_Score}(Q, K, W, \boldsymbol{\beta}) = \text{tril}(QK^\top) - \text{tril}(QW^\top)\left(I + \text{diag}(\boldsymbol{\beta})\text{stril}(WW^\top)\right)^{-1}\text{diag}(\boldsymbol{\beta})\text{stril}(WK^\top).$$

Once we obtain the preconditioned output $Y$ from Algorithm 5, we can compute the final LUCID-PaTH attention output by applying the standard PaTH attention mechanism on $Y$ instead of the original values $V$. This decouples the preconditioning step from the attention computation, enabling efficient implementation:

$$\text{LUCID-PaTH}(Q, K, V, W, \boldsymbol{\beta}) = \left(\text{softmax}\left(\frac{\text{PaTH\_Score}(Q, K, W, \boldsymbol{\beta})}{\sqrt{d}} + \hat{M}\right)\right)Y.$$

The backward pass for LUCID-PaTH requires careful handling of gradients through the block-wise forward-substitution process described in Algorithm 5. Recall that the forward pass solves the lower-triangular system $(I + L)Y = V$, where $L$ is strictly lower-triangular with entries $L_{ij} = \exp(S_{ij})$ for $i > j$. During backpropagation, we must compute gradients with respect to the input values $V$ as well as the PaTH parameters $K_{\text{RN}}, W$, and $\boldsymbol{\beta}$ that determine the attention scores.

The backward pass algorithms follow the structure provided in Appendix Section A.2 while preserving the PaTH implementation structure. Specifically, we maintain the constraint that column blocks must be generated from the diagonal block to the left-most block, as required by (Yang et al., 2025). We present the backward pass algorithms for computing $\nabla_V, \nabla_{K_{\text{RN}}}, \nabla_W$, and $\nabla_{\boldsymbol{\beta}}$ in Algorithm 6, Algorithm 7, and Algorithm 9, respectively.

---

**Algorithm 6** LUCID-PaTH Backward Pass: Computing $\nabla_V$ using Block-wise Backward-Substitution.

---

1: **Input:** $K_{RN} \in \mathbb{R}^{B \times H \times L \times D}, W \in \mathbb{R}^{B \times H \times L \times D}, \boldsymbol{\beta} \in \mathbb{R}^{B \times H \times L}, \nabla_Y \in \mathbb{R}^{B \times H \times L \times D}, \sqrt{d}$
2: **Output:** $\nabla_V \in \mathbb{R}^{B \times H \times L \times D}$
3: // Solve $(I + L)^T \nabla_V = \nabla_Y$ where $L$ is strictly lower triangular
4: // $L^T$ is strictly upper triangular, requiring backward substitution
5: $\nabla_V \leftarrow \nabla_Y.\text{clone}()$
6: $N_{blocks} \leftarrow \text{num\_row\_blocks}$
7: **for** $i = N_{blocks} - 1$ **downto** $0$ **do**
8:    $S_{ii} \leftarrow \text{PaTH\_Score}(K_{RN}, K_{RN}, W, \boldsymbol{\beta}, i, i)/\sqrt{d} - \sqrt{d}$
9:    // Solve diagonal block with transposed upper triangular matrix
10:    $\nabla_V[i] \leftarrow \texttt{torch.linalg.solve\_triangular}(\exp(S_{ii})^T, \nabla_V[i], \text{lower=false}, \text{unitdiagonal=true})$
11:    // Sequential inner loop from diagonal to the left (update earlier blocks)
12:    **for** $k = i - 1$ **downto** $0$ **do**
13:       $S_{ik} \leftarrow \text{PaTH\_Score}(K_{RN}, K_{RN}, W, \boldsymbol{\beta}, i, k)/\sqrt{d} - \sqrt{d}$
14:       $\nabla_V[k] \leftarrow \nabla_V[k] - \exp(S_{ik})^T \nabla_V[i]$
15:    **end for**
16: **end for**
17: **return** $\nabla_V$

---

**Computing $\nabla_V$ via Backward-Substitution.** Algorithm 6 computes the gradient with respect to the input values $V$ by solving the transposed system $(I + L)^T \nabla_V = \nabla_Y$. Since $L$ is strictly lower-triangular, its transpose $L^T$ is strictly upper-triangular, requiring backward-substitution instead of forward-substitution. We initialize $\nabla_V$ with the incoming gradient $\nabla_Y$ (line 5) and process row blocks in reverse order from bottom to top (line 7).

For each row block $i$, we first solve the diagonal block using the transposed upper-triangular matrix $\exp(S_{ii})^T$ (line 10). This step normalizes the gradient for block $i$ by undoing the diagonal scaling applied in the forward pass. The inner loop (lines 12-15) then propagates gradients from block $i$ to all earlier blocks $k < i$ using the relationship $\nabla_V[k] \leftarrow \nabla_V[k] - \exp(S_{ik})^T \nabla_V[i]$. This backward iteration ensures that gradients flow correctly through the forward-substitution dependencies, with each block receiving contributions from all later blocks that depended on it during the forward pass.

**Computing $\nabla_{K_{\mathbf{RN}}}$ through PaTH Scores.** Algorithm 7 computes the gradient with respect to the key matrix $K_{RN}$ by first accumulating gradients with respect to the raw PaTH attention score matrix $A = \text{PaTH\_Score}(K_{RN}, K_{RN}, W, \boldsymbol{\beta})$, then using the PaTH backward pass to obtain $\nabla_{K_{\mathrm{RN}}}$ from $\nabla_A$. The algorithm processes all row blocks (lines 7-19) to compute $\nabla_A$.

For each row block $i$, we handle two types of blocks. The diagonal block $A_{ii}$ involves a gradient through the triangular solver, computed using Algorithm 8 (lines 10). The off-diagonal blocks $A_{ij}$ for $j < i$ are processed in the inner loop (lines 12-18). Here, we first apply the LUCID gradient formula $\nabla_{P_{ij}} = -(\nabla_V[i]Y[j]^T)$ (line 11), which directly computes the gradient with respect to $P = \exp(S)$ without materializing the full attention matrix. We then apply the chain rule to propagate gradients back to the raw score $A$. Since the forward pass computes $S = A/\sqrt{d} - \sqrt{d}$ and then $P = \exp(S)$, the backward chain rule gives $\nabla_A = \nabla_P \odot \exp(S)/\sqrt{d}$ (line 13), where the division by $\sqrt{d}$ accounts for the scaling transformation from $A$ to $S$.

After computing $\nabla_A$ for all blocks, we invoke the PaTH backward routine PaTH\_Backward\_K (line 15) to compute the final gradient $\nabla_{K_{\mathrm{RN}}}$. This routine handles the complex dependencies in the PaTH score formula. The abstraction of this step allows us to leverage the existing PaTH implementation while focusing on the LUCID-specific gradient computations.

**Gradient through Diagonal Block Triangular Solver.** Algorithm 8 is a helper routine that computes the gradient with respect to the diagonal block score $A_{ii}$ through the triangular solver used in the forward pass. Recall that in the forward pass, we solve the lower-triangular system $(I + L_{ii})Y[i] = V_{\text{input}}[i]$ where $L_{ii} = \exp(S_{ii}) \odot M_{\text{stril}}$ and $S_{ii} = A_{ii}/\sqrt{d} - \sqrt{d}$.

The gradient computation applies the LUCID formula for the triangular solver: $\nabla_{P_{ii}} = -(\nabla_V[i]Y[i]^T) \odot M_{\text{stril}}$ (line 6), where the mask $M_{\text{stril}}$ restricts the gradient to the strictly lower-triangular part since the diagonal is implicitly unit-valued. We then apply the chain rule (lines 8-9) to propagate the gradient from $P_{ii} = \exp(S_{ii})$ back to the raw score $A_{ii}$, giving $\nabla_{A_{ii}} = \nabla_{P_{ii}} \odot \exp(S_{ii})/\sqrt{d}$. This follows the same backward chain rule as the off-diagonal blocks, accounting for the

---

**Algorithm 7** LUCID-PaTH Backward Pass: Computing $\nabla_{K_{\text{RN}}}$.

---

1: **Input:** $K_{RN} \in \mathbb{R}^{B \times H \times L \times D}, W \in \mathbb{R}^{B \times H \times L \times D}, \boldsymbol{\beta} \in \mathbb{R}^{B \times H \times L}, \nabla_V \in \mathbb{R}^{B \times H \times L \times D}, Y \in \mathbb{R}^{B \times H \times L \times D}, \sqrt{d}$
2: **Output:** $\nabla_{K_{\text{RN}}} \in \mathbb{R}^{B \times H \times L \times D}$
3: // Note: $\nabla_V$ is obtained from Algorithm 6
4: $\nabla_{K_{\text{RN}}} \leftarrow \mathbf{0}$
5: $N_{blocks} \leftarrow$ num_row_blocks
6: // Process all blocks to compute $\nabla_A$ where $A = \text{PaTH\_Score}(K_{RN}, K_{RN}, W, \boldsymbol{\beta})$
7: **for** $i = 0$ **to** $N_{blocks} - 1$ **do**
8:     // Diagonal block: gradient through triangular solver (see Algorithm 8)
9:     $A_{ii} \leftarrow \text{PaTH\_Score}(K_{RN}, K_{RN}, W, \boldsymbol{\beta}, i, i)$
10:     $\nabla_{A_{ii}} \leftarrow \text{TriangularSolveGrad}(A_{ii}, \nabla_V[i], Y[i], \sqrt{d})$
11:     // Off-diagonal blocks: use LUCID formula $\nabla_P = -(\nabla_V Y^T) \odot M$
12:     **for** $j = i - 1$ **downto** $0$ **do**
13:         $A_{ij} \leftarrow \text{PaTH\_Score}(K_{RN}, K_{RN}, W, \boldsymbol{\beta}, i, j)$
14:         $S_{ij} \leftarrow A_{ij}/\sqrt{d} - \sqrt{d}$
15:         $\nabla_{P_{ij}} \leftarrow -(\nabla_V[i]Y[j]^T)$
16:         // Chain rule: $P = \exp(S), S = A/\sqrt{d} - \sqrt{d}$
17:         $\nabla_{A_{ij}} \leftarrow \nabla_{P_{ij}} \odot \exp(S_{ij})/\sqrt{d}$
18:     **end for**
19: **end for**
20: // Use PaTH backward to compute $\nabla_{K_{\text{RN}}}$ from $\nabla_A$
21: $\nabla_{K_{\text{RN}}} \leftarrow \text{PaTH\_Backward\_K}(\nabla_A, K_{RN}, W, \boldsymbol{\beta})$
22: **return** $\nabla_{K_{\text{RN}}}$

---

**Algorithm 8** TriangularSolveGrad: Gradient through diagonal block triangular solver.

---

1: **Input:** $A_{ii} \in \mathbb{R}^{B \times H \times BS \times BS}, \nabla_V[i] \in \mathbb{R}^{B \times H \times BS \times D}, Y[i] \in \mathbb{R}^{B \times H \times BS \times D}, \sqrt{d}$
2: **Output:** $\nabla_{A_{ii}} \in \mathbb{R}^{B \times H \times BS \times BS}$
3: // Forward: $S_{ii} = A_{ii}/\sqrt{d} - \sqrt{d}, L_{ii} = \exp(S_{ii}) \odot M_{\text{stril}}$
4: // Forward: $(I + L_{ii})Y[i] = V_{\text{input}}[i]$ solved via triangular solver
5: // LUCID gradient formula for triangular solve
6: $\nabla_{P_{ii}} \leftarrow -(\nabla_V[i]Y[i]^T) \odot M_{\text{stril}}$
7: // Chain rule: $P = \exp(S), S = A/\sqrt{d} - \sqrt{d}$
8: $S_{ii} \leftarrow A_{ii}/\sqrt{d} - \sqrt{d}$
9: $\nabla_{A_{ii}} \leftarrow \nabla_{P_{ii}} \odot \exp(S_{ii})/\sqrt{d}$
10: **return** $\nabla_{A_{ii}}$

---

exponential and scaling transformations.

**Computing $\nabla_W$ and $\nabla_{\boldsymbol{\beta}}$ through PaTH Scores.** Algorithm 9 computes the gradients with respect to the PaTH-specific parameters $W$ and $\boldsymbol{\beta}$ using a structure similar to Algorithm 7. The algorithm first computes $\nabla_A$ by processing all blocks (lines 8-20), handling diagonal blocks through Algorithm 8 (lines 11) and off-diagonal blocks using the LUCID gradient formula (lines 13-19). The gradient computation for each block is identical to that in Algorithm 7, since both algorithms require $\nabla_A$ as an intermediate result.

The key difference lies in the final step (line 22), where we invoke PaTH_Backward_W_Beta to compute $\nabla_W$ and $\nabla_{\boldsymbol{\beta}}$ from $\nabla_A$. This routine handles the complex dependencies on $W$ and $\boldsymbol{\beta}$ in the PaTH score formula, particularly in the correction term $\text{tril}(K_{\text{RN}}W^\top)\left(I + \text{diag}(\boldsymbol{\beta})\text{stril}(WW^\top)\right)^{-1}\text{diag}(\boldsymbol{\beta})\text{stril}(WK_{\text{RN}}^\top)$. By abstracting these PaTH-specific computations, we maintain a clean separation between the LUCID preconditioning logic and the PaTH positional encoding mechanism.

---

**Algorithm 9** LUCID-PaTH Backward Pass: Computing $\nabla_W$ and $\nabla_{\boldsymbol{\beta}}$.

---

1: **Input:** $K_{RN} \in \mathbb{R}^{B \times H \times L \times D}, W \in \mathbb{R}^{B \times H \times L \times D}, \boldsymbol{\beta} \in \mathbb{R}^{B \times H \times L}, \nabla_V \in \mathbb{R}^{B \times H \times L \times D}, Y \in \mathbb{R}^{B \times H \times L \times D}, \sqrt{d}$
2: **Output:** $\nabla_W \in \mathbb{R}^{B \times H \times L \times D}, \nabla_{\boldsymbol{\beta}} \in \mathbb{R}^{B \times H \times L}$
3: // Note: $\nabla_V$ is obtained from Algorithm 6
4: $\nabla_W \leftarrow \mathbf{0}$
5: $\nabla_{\boldsymbol{\beta}} \leftarrow \mathbf{0}$
6: $N_{blocks} \leftarrow \text{num\_row\_blocks}$
7: // Process all blocks to compute $\nabla_A$ where $A = \text{PaTH\_Score}(K_{RN}, K_{RN}, W, \boldsymbol{\beta})$
8: **for** $i = 0$ **to** $N_{blocks} - 1$ **do**
9:     // Diagonal block: gradient through triangular solver (see Algorithm 8)
10:     $A_{ii} \leftarrow \text{PaTH\_Score}(K_{RN}, K_{RN}, W, \boldsymbol{\beta}, i, i)$
11:     $\nabla_{A_{ii}} \leftarrow \text{TriangularSolveGrad}(A_{ii}, \nabla_V[i], Y[i], \sqrt{d})$
12:     // Off-diagonal blocks: use LUCID formula $\nabla_P = -(\nabla_V Y^T) \odot M$
13:     **for** $j = i - 1$ **downto** $0$ **do**
14:         $A_{ij} \leftarrow \text{PaTH\_Score}(K_{RN}, K_{RN}, W, \boldsymbol{\beta}, i, j)$
15:         $S_{ij} \leftarrow A_{ij}/\sqrt{d} - \sqrt{d}$
16:         $\nabla_{P_{ij}} \leftarrow -(\nabla_V[i]Y[j]^T)$
17:         // Chain rule: $P = \exp(S), S = A/\sqrt{d} - \sqrt{d}$
18:         $\nabla_{A_{ij}} \leftarrow \nabla_{P_{ij}} \odot \exp(S_{ij})/\sqrt{d}$
19:     **end for**
20: **end for**
21: // Use PaTH backward to compute $\nabla_W$ and $\nabla_{\boldsymbol{\beta}}$ from $\nabla_A$
22: $\nabla_W, \nabla_{\boldsymbol{\beta}} \leftarrow \text{PaTH\_Backward\_W\_Beta}(\nabla_A, K_{RN}, W, \boldsymbol{\beta})$
23: **return** $\nabla_W, \nabla_{\boldsymbol{\beta}}$

---

## A.4. Relation to DeltaNet

As established in Section 2.1, LUCID can be viewed as DeltaNet operating in the infinite-dimensional RKHS induced by the exponential kernel. Here we make this connection precise and highlight the key differences.

DeltaNet (Yang et al., 2024b) proposes a preconditioned linear attention mechanism:

$$\left(M \circ QK^\top\right)\left(I_N + \text{stril}\left(\text{diag}(\boldsymbol{\beta})KK^\top\right)\right)^{-1}\text{diag}(\boldsymbol{\beta})V,$$

where $\boldsymbol{\beta} = (\beta_1, \ldots, \beta_N)$ are learned per-token scaling parameters. Setting $\boldsymbol{\beta} = 1$ and introducing the exponential kernel feature map $\phi$ recovers the LUCID formulation.

**Complexity Comparison.** DeltaNet achieves $\mathcal{O}(Nd^2)$ time and $\mathcal{O}(d^2)$ memory via its recurrent formulation, making it suitable for very long sequences. LUCID, in contrast, retains $\mathcal{O}(N^2d)$ complexity like standard softmax attention. This is intentional: LUCID is designed as a drop-in enhancement for full attention layers, not a replacement for efficient attention. In hybrid architectures that combine efficient layers (e.g., sliding window, SSMs) with global attention layers, LUCID improves the precision of the global layers where retrieval quality is paramount.

**Memory Update Capability.** A key feature of DeltaNet is its ability to update values associated with keys—when the same key appears multiple times in a sequence, DeltaNet can overwrite the old value with the new one. This "erase-then-write" mechanism, derived from the delta rule, is essential for binding context-specific values to keys. LUCID inherits this capability through its derivation from the same delta rule in RKHS (Section 2.1).

**Key Difference: Feature Space Dimensionality.** The fundamental distinction lies in the feature space:

- **DeltaNet**: Uses the identity map $\phi(x) = x$, operating in the $d$-dimensional token space. The preconditioner $(I + \text{stril}(KK^\top))^{-1}$ decorrelates keys in this finite-dimensional space.

- **LUCID**: Uses the exponential kernel feature map $\phi : \mathbb{R}^d \to \mathcal{H}$, operating in an infinite-dimensional RKHS. The preconditioner $(M \circ \exp(KK^\top))^{-1}$ decorrelates keys in this richer space.

*Table 5.* Performance comparison of different variants of LUCID Attention with different positional encoding for MNIAH task at 2048 sequence length. The top half has RoPE.

| Variant | Number of Needles | | | | |
|---|---|---|---|---|---|
| | **2** | **4** | **6** | **8** | **10** |
| **key norm** ($\beta = 1$) | **76.6** | **55.8** | **43.6** | **33.8** | 25.2 |
| **QK-Norm** | 75.4 | 51.4 | 37.2 | 32.0 | 20.4 |
| **max norm** | 74.2 | 48.8 | 37.2 | 31.4 | 19.6 |
| **key norm (learn $\beta$)** | 73.8 | 48.0 | 36.8 | 31.8 | **26.8** |
| **NoPE key norm** ($\beta = 1$) | 60.6 | 37.6 | 30.4 | 22.0 | 19.2 |
| **NoPE QK norm** | **87.0** | **69.4** | **61.6** | **50.6** | **43.4** |
| **ALiBi key norm** ($\beta = 1$) | 74.6 | 53.2 | 43.6 | 32.2 | 29.8 |
| **ALiBi QK norm** | 59.0 | 30.8 | 24.2 | 17.6 | 14.2 |

**Why Infinite Dimensions Matter.** In the exponential kernel RKHS, a crucial property holds: $\langle \phi(\mathbf{k}_i), \phi(\mathbf{k}_j) \rangle = \exp(\mathbf{k}_i^\top \mathbf{k}_j) > 0$ for all $i, j$. This means keys are *never orthogonal* in the RKHS feature space, even when they are orthogonal in token space. Consequently, the LUCID preconditioner always provides non-trivial correction, whereas DeltaNet's correction vanishes when keys happen to be orthogonal in the $d$-dimensional space. This richer geometry allows LUCID to capture and remove interference patterns that DeltaNet cannot detect, leading to more precise retrieval. In our experiments, we did not find performance gains from learning $\beta$.

> **LUCID = DeltaNet in RKHS** The key difference is dimensionality: DeltaNet operates in finite $d$-dimensional token space where keys can be orthogonal; LUCID operates in infinite-dimensional RKHS where $\exp(\mathbf{k}_i^\top \mathbf{k}_j) > 0$ always, enabling complete decorrelation.

### A.5. Ablations

**Reduced Head Size.** We next examine the effect of reducing the head dimension while keeping the total model dimension constant. Specifically, we fix the sequence length to 4096 from the experiment in Section A.5 and reduce the per-head size from 64 to 16, increasing the number of heads proportionally from 32 to 128.

Under this setup, LUCID attention achieves a 0.1 improvement in validation loss over the Transformer baseline for headsize 16. This confirms our intuition that LUCID is most effective when the attention representation is narrow relative to the sequence length.

**RoPE.** Rotary Position Embeddings (RoPE) introduce relative position information by applying structured rotation matrices to the query and key vectors. We have found that the loss difference between LUCID (2.42) and standard attention (2.54) increases to 0.12 when the RoPE is removed in sequence length 32k setting (Figure 7). We conjecture that RoPE structures the keys and queries there by conditioning them for better retrieval. Absence of RoPE, alternatively called NoPE, is used in Llama 4 (Meta AI, 2025).

**LUCID Variants.** We test different LUCID variants along with NoPE and ALiBi (Press et al., 2022) positional encoding on MNIAH task for different number of needles for 2048 sequence length obtained by experimenting with $\exp$ stabilization. The results are summarized in Table 5. LUCID with NoPE and QK norm provides the best MNIAH performance and LUCID with Alibi and with key norm ($\beta = 1$) is competitive at 10 needles. This demonstrates the compatibility of LUCID with other positional encoding schemes.

**Increasing Pretraining Context Length** We also conduct some log-perplexity based evaluations at larger scale by training a 16 layer model with model dimension 2048, feed-forward dimension 4096, vocabulary size 256,000, and $\beta_2 = 0.95$ for Adam. We trained the models on sequence lengths 4096 and 32,768. For the 4096-token setup, we train with a batch size of 1M tokens per iteration for 80k steps (80B tokens). For the 32K-token setting, we use 2M tokens per iteration for 25k steps (50B tokens). We evaluate how LUCID performs as sequence length increases while holding attention head size constant. At 4096 context length, LUCID improves training loss by 0.012 over the Transformer baseline after 80B tokens. At 32,768

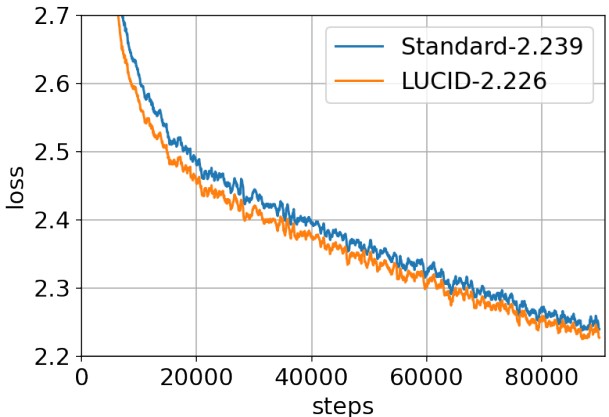 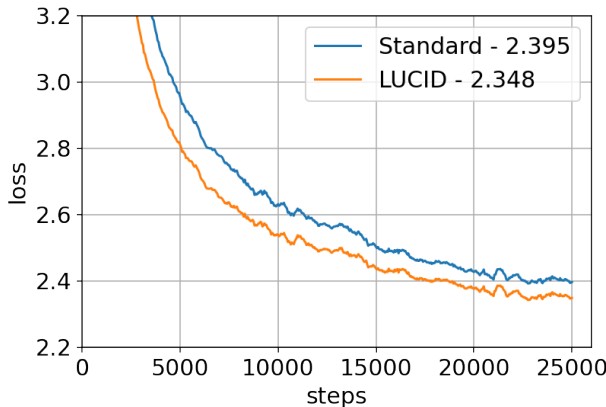

*Figure 7.* Training loss vs. steps for Standard Attention vs. LUCID Attention at different context lengths. LUCID achieves lower validation loss at both 4096 (left) and 32768 (right) sequence lengths.

*Table 6.* Performance comparison of Standard Attention and LUCID Attention models for SNIAH and MNIAH tasks across different sequence lengths after fine-tuning.

| Task | Model | Sequence Length | | | | | | |
|---|---|---|---|---|---|---|---|---|
| | | **2K** | **4K** | **6K** | **8K** | **16K** | **32K** | **64K** |
| **SNIAH** | **Standard** | 55.9 | 47.5 | 32.5 | 19.3 | 2.0 | 0.0 | 2.0 |
| | **LUCID** | 62.2 | 68.6 | 68.2 | 50.2 | 15.0 | 1.0 | 2.0 |
| **MNIAH** | **Standard** | 33.6 | 37.4 | 26.0 | 23.2 | 4.0 | 4.0 | 2.0 |
| | **LUCID** | 51.6 | 51.4 | 40.4 | 33.2 | 14.0 | 12.0 | 2.0 |

context length, the gain increases to 0.048 under the same architecture and optimization settings. These results support our hypothesis: for a fixed representation size, the need to precondition grows with sequence length.

**Compute Ablation**    A natural question is whether the 0-6% training overhead of LUCID could be better utilized by simply training the baseline model for longer. To address this, we conducted an ablation study where we increased the training compute for Standard Attention by ~10% additional steps during 64K sequence length continual pretraining, matching LUCID's total compute budget. As shown in Table 7, training the baseline longer does not yield equivalent performance gains—LUCID significantly outperforms the compute-matched baseline, particularly in the critical 8K–16K context window where standard attention begins to fail. This demonstrates that LUCID's improvements stem from its architectural design rather than merely additional compute.

*Table 7.* Compute ablation: Standard Attention trained for +10% additional steps vs. LUCID on needle-in-a-haystack tasks. Extra training compute does not close the performance gap.

| Task | Seq Len | Standard | Standard (+10%) | LUCID |
|---|---|---|---|---|
| SNIAH | 4096 | 0.475 | 0.467 | **0.686** |
| | 8192 | 0.193 | 0.193 | **0.502** |
| | 16384 | 0.020 | 0.020 | **0.150** |
| MNIAH | 8192 | 0.232 | 0.234 | **0.332** |
| | 16384 | 0.040 | 0.060 | **0.140** |
| | 32768 | 0.040 | 0.040 | **0.120** |

## A.6. LongBench and SCROLLS Task Details

We evaluate on six tasks spanning three categories:

- **Multi-Document QA**: *2WikiMQA* (Ho et al., 2020) requires multi-hop reasoning with up to 5-hop chains over Wikipedia passages. *HotpotQA* (Yang et al., 2018) tests 2-hop QA mixed with numerous distracting passages.

- **Single-Document QA**: *MultifieldQA* (Bai et al., 2023) covers diverse domains including legal, government, and academic documents. *Qasper* (Dasigi et al., 2021) (8K context) requires information-seeking QA over full-text NLP papers.

- **Summarization**: *QMSum* (Zhong et al., 2021) (32K context) evaluates query-based summarization of long meeting transcripts.

**Finetuning Setup.** To ensure rigorous evaluation, we adopted the following finetuning protocol:

- **LongBench Tasks**: We finetuned on a combination of 50% of the 2WikiMQA training data and 50% of the MultifieldQA training data. We then evaluated on the remaining halves of 2WikiMQA and MultifieldQA, and the full HotpotQA dataset.

- **SCROLLS Tasks**: For QMSum and Qasper, we finetuned on their respective training splits and evaluated on their validation splits.

### A.7. LUCID Scaling

This section details the scalability and performance of the LUCID attention mechanism across 500M, 1B, and 2B parameter scales. Our empirical results indicate that LUCID not only maintains competitive performance but also demonstrates a favorable scaling trajectory in both pre-training efficiency and downstream task performance.

### A.8. Efficiency and Pre-training Convergence

A primary advantage of the LUCID architecture is its consistent ability to achieve lower final pre-training cross-entropy loss compared to the Softmax baseline across all tested model sizes. As summarized in Table 8, the performance advantage over Softmax scales positively with model capacity. Specifically, the reduction in loss ($\Delta$) improves from -0.0196 at the 500M scale to -0.0276 at the 2B scale. This widening gap suggests that the LUCID preconditioner provides increasing benefits as parameter count scales.

*Table 8.* Final Pre-training Loss comparison (lower is better).

| Model Size | LUCID | Softmax | Delta |
| --- | --- | --- | --- |
| 500M | **2.5915** | 2.6111 | -0.0196 |
| 1B | **2.4293** | 2.4522 | -0.0229 |
| 2B | **2.2930** | 2.3206 | -0.0276 |

### A.9. Scaling of Zero-Shot Capabilities

Zero-shot evaluations across 12 benchmarks, including SuperGLUE (Wang et al., 2020), TinyBenchmarks (Maia Polo et al., 2024), ARC (Clark et al., 2018), and MMLU (Hendrycks et al., 2021), indicate that LUCID effectively captures downstream knowledge across various domains. LUCID demonstrates significant gains as model size increases; notably, on the WSC benchmark (Wang et al., 2020), performance improves from 0.3654 at 500M to 0.6442 at 2B, substantially outpacing the growth of the baseline. Furthermore, LUCID maintains a lead in foundational reasoning tasks such as ARC Challenge and ARC Easy (Clark et al., 2018), reaching 0.2944 and 0.5560 respectively at the largest tested scale. Complete zero-shot results are detailed in Table 9.

### A.10. Robustness in In-Context Learning

The 5-shot in-context learning results, summarized in Table 10, demonstrate that the LUCID architecture successfully preserves the model's ability to utilize demonstrations effectively. At the 2B parameter level, LUCID achieves robust performance across MMLU (Hendrycks et al., 2021) and ARC (Clark et al., 2018) benchmarks, showing a marginal lead in tasks such as ARC Easy (0.6376 vs. 0.6317) and MMLU (0.2606 vs. 0.2569). These results suggest that the LUCID preconditioner enhances fundamental training stability without compromising emergent few-shot capabilities.

*Table 9.* 0-shot Benchmark Accuracy (L denotes LUCID, S denotes Softmax).

| Benchmark | 500M L | 500M S | 1B L | 1B S | 2B L | 2B S |
|---|---|---|---|---|---|---|
| boolq (acc) (Wang et al., 2020) | **0.6199** | 0.5963 | 0.6190 | **0.6220** | **0.5985** | 0.5972 |
| cb (acc) (Wang et al., 2020) | **0.4643** | 0.3929 | **0.5893** | 0.3571 | **0.4643** | 0.3393 |
| multirc (acc) (Wang et al., 2020) | 0.5708 | **0.5720** | **0.5716** | 0.5703 | **0.5714** | 0.5559 |
| record (f1) (Wang et al., 2020) | **0.5754** | **0.5754** | **0.6164** | 0.6131 | **0.6564** | 0.6519 |
| sglue_rte (acc) (Wang et al., 2020) | **0.5379** | 0.5054 | 0.5162 | **0.5487** | **0.4946** | 0.4729 |
| wsc (acc) (Wang et al., 2020) | 0.3654 | **0.3846** | **0.4135** | 0.3654 | **0.6442** | 0.4135 |
| tinyHellaswag (Maia Polo et al., 2024) | 0.2280 | **0.2879** | **0.3147** | 0.2651 | **0.3784** | 0.3302 |
| tinyTruthfulQA (Maia Polo et al., 2024) | **0.4513** | 0.4394 | 0.4168 | **0.4264** | **0.4113** | 0.3893 |
| tinyWinogrande (Maia Polo et al., 2024) | **0.4821** | 0.4351 | 0.4413 | **0.5630** | **0.4837** | 0.4522 |
| ARC Challenge (Clark et al., 2018) | **0.2509** | 0.2474 | **0.2747** | 0.2611 | **0.2944** | 0.2884 |
| ARC Easy (Clark et al., 2018) | **0.4735** | 0.4465 | **0.5084** | 0.4962 | **0.5560** | 0.5299 |
| mmlu (Hendrycks et al., 2021) | **0.2313** | 0.2300 | **0.2324** | 0.2295 | 0.2443 | **0.2491** |

*Table 10.* 5-shot In-Context Learning Performance (L denotes LUCID, S denotes Softmax).

| Benchmark | 500M L | 500M S | 1B L | 1B S | 2B L | 2B S |
|---|---|---|---|---|---|---|
| ARC Challenge (Clark et al., 2018) | 0.2526 | **0.2594** | **0.2875** | 0.2867 | 0.3217 | **0.3234** |
| ARC Easy (Clark et al., 2018) | **0.5072** | 0.4958 | **0.5741** | 0.5648 | **0.6376** | 0.6317 |
| mmlu (Hendrycks et al., 2021) | 0.2559 | **0.2578** | **0.2548** | 0.2516 | **0.2606** | 0.2569 |

