# OpenReview forum: "LUCID: Attention with Preconditioned Representations"
_ICML.cc/2026/Conference — ICML 2026 regular_

### Official Review · Reviewer_9BN1 · 2026-03-04

**Soundness:** 3
**Presentation:** 3
**Significance:** 3
**Originality:** 3
**Overall Recommendation:** 4
**Confidence:** 4

**Summary:**

The authors introduce LUCID Attention, a new attention variant that aims to decorrelate keys (in the exponential-kernel RKHS), and thus reduce attention noise, allowing the queries to focus more accurately on important keys and providing more precise attention distributions.
They do that by (1) using the causal pairwise similarity matrix of K with itself (via exp(KKᵀ)) as a preconditioner, and (2) preconditioning the values by applying the inverse of this matrix (implemented as a triangular solve), before combining with the usual exp(QKᵀ) attention weights.
Authors show results improvement over other attention variants over BABILong, NIAH, SCROLLS and LongBench, and show that this can be piggybacked on regular attention operations with only a small computation overhead.

**Compliance With Llm Reviewing Policy:**

Affirmed.

**Final Justification:**

The authors provided results on compatibility with NoPE, ALiBi, and QK-norm, added diagnostics supporting the claimed decorrelation effect, and reported efficiency numbers.
While some evaluation details could still be more exhaustive, I find the core contribution technically solid and useful.
I therefore maintain my Weak Accept recommendation.

**Key Questions For Authors:**

As I mentioned before,
1.  How does LUCID interact with other common attention modifications beyond PaTH—e.g., ALiBi, QK-norm on/off, and alternative positional schemes (RoPE variants / NoPE)—especially ones that may change key correlations and long-context recall?
2. Can you provide more thorough, reproducible efficiency results—e.g., plots/tables for training and inference throughput and peak memory—across sequence lengths and model sizes, and specify the exact hardware/software settings?
3. If possible, I’d love to see additional interpretability or diagnostic results that directly test the proposed mechanism. For example, before/after measurements of the key–key similarity matrix or related correlation metrics, so the reader can verify the intended "decorrelation" effect.

**Limitations:**

The authors mention one limitation: extending LUCID to bidirectional settings removes the triangular structure and requires a more expensive linear solve. My comments mainly concern additional experiments rather than concrete limitations, so I have nothing further to suggest.

**Strengths And Weaknesses:**

** Strengths**
1. Clearly written and easy to follow.
2. Simple method: an elegant mathematical framing/derivation (RKHS view and preconditioning interpretation) that reduces long-context attention noise from correlated keys via RKHS-motivated preconditioning, improving retrieval focus.
3. Modest overhead (triangular solve) while keeping the same O(N^2 d) scaling as full attention.
4. Strong empirical gains across long-context benchmarks (BABILong, NIAH, SCROLLS, LongBench), including multi-needle retrieval

**Weaknesses**
1. Ablations are mostly confined to the appendix; it’s unclear how LUCID interacts with other common attention changes beyond PaTH (e.g., ALiBi, QK-norm on/off, alternative positional schemes) that can affect key correlations and long-context recall.
2. Throughput/efficiency reporting is not very thorough: training/inference speed and memory impact are described briefly rather than with detailed, reproducible plots.
3. They do not directly validate the proposed “key decorrelation” mechanism with before/after measurements (e.g., covariance/similarity heatmaps or quantitative correlation metrics for KK^\top or \exp(KK^\top)). Evidence is mainly indirect (condition number proxies and downstream retrieval performance).

---

> ### Author Rebuttal · Authors · 2026-03-31
>
> We appreciate the useful comments provided by the reviewer and have addressed some concerns below.
>
> > How does LUCID interact with other common attention modifications beyond PaTH—e.g., ALiBi, QK-norm on/off, and alternative positional schemes (RoPE variants / NoPE)—especially ones that may change key correlations and long-context recall?
>
> As requested by the reviewer, we did MNIAH evaluation of NoPE and Alibi with and without QK-norm and the results are as follows.
>
> | Model | 2 | 4 | 6 | 8 | 10 |
> |---|---|---|---|---|---|
> | Lucid Nope exp K norm | 0.606 | 0.376 | 0.304 | 0.22 | 0.192 |
> | Lucid Nope QK norm | 0.87 | 0.694 | 0.616 | 0.506 | 0.434 |
> | Lucid Alibi exp K norm | 0.746 | 0.532 | 0.436 | 0.322 | 0.298 |
> | Lucid Alibi QK norm | 0.59 | 0.308 | 0.242 | 0.176 | 0.142 |
>
> LUCID with NoPE and QK norm provides the best MNIAH performance and LUCID with Alibi and without QK norm is competitive at 10 needles. This demonstrates the compatibility of LUCID with other positional encoding schemes.
>
> > If possible, I’d love to see additional interpretability or diagnostic results that directly test the proposed mechanism. For example, before/after measurements of the key–key similarity matrix or related correlation metrics, so the reader can verify the intended "decorrelation" effect.
>
> We analyzed the keys of LUCID and standard softmax and plotted the distribution of the key-key alignment measured by the absolute value of cosine of angle between the keys, where we find LUCID to have smaller correlations than softmax.
>
> **Mean Key Correlation**
>
> | Model | 512 | 1024 | 2048 | 4096 | 8192 | 16384 |
> |---|---|---|---|---|---|---|
> | Softmax | 0.5611 | 0.5213 | 0.4975 | 0.4232 | 0.3095 | 0.2387 |
> | Lucid | 0.2637 | 0.2062 | 0.1711 | 0.1345 | 0.1188 | 0.1105 |
>
> > Can you provide more thorough, reproducible efficiency results—e.g., plots/tables for training and inference throughput and peak memory—across sequence lengths and model sizes, and specify the exact hardware/software settings?
>
>
> We trained the models on GH200 GPUs available. The training and inference throughput is provided below.
>
> | Length | LUCID Inference | Softmax Inference | Inference Overhead | LUCID Train Throughput | Softmax Train Throughput | Train Overhead |
> |---|---|---|---|---|---|---|
> | 4K | 122K tok/s | 129K tok/s | 6% | 23K tok/s | 24K tok/s | 6% |
> | 8K | 118K tok/s | 123K tok/s | 5% | 30K tok/s | 31K tok/s | 6% |
> | 16K | 103K tok/s | 108K tok/s | 4% | 25K tok/s | 27K tok/s | 6% |
> | 32K | 78K tok/s | 82K tok/s | 5% | 18K tok/s | 19K tok/s | 6% |
> | 64K | 52K tok/s | 55K tok/s | 4% | 12K tok/s | 12K tok/s | 5% |
> | 128K | 31K tok/s | 32K tok/s | 4% | 7K tok/s | 7K tok/s | 6% |
>
> The table reports inference (prefill only) - Softmax Infer and training throughput (in tokens/second) for LUCID and standard Softmax attention across sequence lengths from 4K to 128K, using a head dimension of 128 and a GQA ratio of 2:1. Throughput peaks around 4K–8K tokens where GPU utilization is optimal, then decreases at longer sequences as memory bandwidth becomes the bottleneck. To achieve the above, we implemented LUCID's forward substitution by modifying the Flash-Attention 3 CuTe DSL kernel. We found that this implementation utilizes same memory as standard attention, additionally yielding substantial speedups over the naive torch.linalg.solve_triangular implementation: 18× at 8K (118K vs 7K tok/s), 39× at 32K (78K vs 2K tok/s), and 61× at 64K (53K vs 865 tok/s). We will make the kernel implementation and benchmarking exhaustive in the final revision.

---

> > ### Author Rebuttal · Reviewer_9BN1 · 2026-04-01
> >
> > My concerns have been addressed. I thank the authors for the answers. I keep my score.

---

### Official Review · Reviewer_KuxF · 2026-03-08

**Soundness:** 3
**Presentation:** 3
**Significance:** 3
**Originality:** 2
**Overall Recommendation:** 4
**Confidence:** 3

**Summary:**

This paper proposes LUCID Attention, a modification of softmax attention designed to improve retrieval in long-context settings. The key idea is to interpret attention as a retrieval process in a kernel feature space and derive a correction based on minimizing a quadratic retrieval error objective. This leads to a preconditioning mechanism based on exponentiated key-key similarities, which decorrelates keys in the RKHS feature space. As a result, the method aims to achieve sharper retrieval behavior comparable to low-temperature softmax while maintaining non-vanishing gradients for stable learning. Experiments on multi-needle retrieval tasks and long-context reasoning benchmarks (e.g., BABILong) show improved performance with modest training overhead.

**Compliance With Llm Reviewing Policy:**

Affirmed.

**Final Justification:**

My concerns are not fully addressed. But as the paper has good quality in general, I vote for acceptance.

**Key Questions For Authors:**

Questions for the Authors

1. The analysis mainly focuses on retrieval tasks. How does the proposed attention mechanism affect long-horizon autoregressive generation behavior in practice?

2. Since the method requires exponentiated key-key similarities and triangular solves, how robust is the method under mixed-precision training or inference?

3. Can the authors clarify more explicitly how LUCID differs from DeltaNet beyond the RKHS interpretation?

**Limitations:**

yes

**Strengths And Weaknesses:**

Strengths
- The paper is clearly written and the derivation is relatively easy to follow.
- The method is well motivated from the perspective of retrieval in kernel feature space.
- The theoretical discussion connecting attention updates to optimization objectives provides useful insight.
- The implementation details and computational overhead analysis are presented clearly.
- Experiments on long-context retrieval benchmarks are reasonably designed and show consistent improvements.

Weaknesses
- The conceptual novelty compared with DeltaNet appears limited. The paper claims that LUCID can be viewed as DeltaNet generalized to RKHS, but the practical differences and advantages over DeltaNet are not analyzed in sufficient depth. A clearer comparison would strengthen the contribution.
- The terminology "attentional noise" may be somewhat misleading. The proposed mechanism primarily decorrelates keys through a preconditioning matrix rather than explicitly suppressing noise. It might be clearer to describe the effect as correcting key correlations rather than removing noise.
- Some theoretical arguments rely on ignoring the softmax denominator when interpreting attention updates as gradient descent. However, in practice the denominator may partially mitigate the accumulation of interference. It would be helpful to clarify under what conditions this approximation is valid.
- The method involves exponentiation of key-key similarities and solving triangular systems. It is unclear how numerically stable the method is under low-precision training or inference (e.g., fp16 or fp8), which are common in modern large-scale LLM deployments.

---

> ### Author Rebuttal · Authors · 2026-03-31
>
> We thank the reviewer for their review and would like to address the questions raised.
>
>
> > Can the authors clarify more explicitly how LUCID differs from DeltaNet beyond the RKHS interpretation?
>
> Beyond what's mentioned in Appendix A.4, LUCID and DeltaNet differ in three fundamental respects beyond the RKHS interpretation. First, DeltaNet uses the identity map $\phi(x) = x$ in finite d-dimensional space, where orthogonal keys cause the preconditioner correction to vanish; LUCID's exponential kernel guarantees $exp(k_i^\top k_j) > 0$ always, ensuring non-trivial correction regardless of key geometry. Second, DeltaNet replaces softmax entirely with linear attention at $O(Nd^2)$ complexity, whereas LUCID retains softmax and applies the preconditioner multiplicatively to the values. Third, DeltaNet employs learned per-token scaling $\beta$; LUCID fixes $\beta = 1$ with RMS key normalization, which we find empirically superior (Table 5).
>
> > Since the method requires exponentiated key-key similarities and triangular solves, how robust is the method under mixed-precision training or inference?
>
> Robustness under lower precision dtypes, like bfloat16 or fp8, is indeed useful for LLM production. The Torch triangular solver doesn’t natively support lower precision than fp32. So, we wrote a kernel for LUCID’s forward substitution/triangular solver by modifying Flash-Attention 3 CUTLASS DSL kernel which has these features built in to address this and will make it exhaustive in our final revision (Please see our response to 9BN1 to find more details). As a byproduct, this yields substantial speedups over the naive torch.linalg.solve_triangular implementation: 18× at 8K (118K vs 7K tok/s), 39× at 32K (78K vs 2K tok/s), and 61× at 64K (53K vs 865 tok/s).

---

> > ### Author Rebuttal · Reviewer_KuxF · 2026-04-02
> >
> > The authors did not fully address some of my concerns.
> > Nevertheless, I find the paper to be of good quality and potential significance, and I therefore support acceptance.

---

### Official Review · Reviewer_xRQn · 2026-03-10

**Soundness:** 3
**Presentation:** 3
**Significance:** 2
**Originality:** 3
**Overall Recommendation:** 5
**Confidence:** 4

**Summary:**

This paper addresses a problem in the standard attention mechanism in which, for longer contexts, the attention score distribution becomes flatter (entropy becomes larger), which introduces noise to the backpropagation of training signals (gradients). Meanwhile, using a smaller temperature in standard attention causes gradients to vanish. The authors propose LUCID, a novel attention mechanism with a preconditioner based on the Delta Rule to avoid the vanishing gradient problem. Empirical results show that LUCID outperforms existing attention mechanisms in retrieval-related tasks.

**Compliance With Llm Reviewing Policy:**

Affirmed.

**Final Justification:**

This paper proposes a novel modification to the softmax attention mechanism that improves its long-context abilities by better balancing attention entropy and gradient signals. The experimental results are rather convincing, but are limited to long-context abilities. My original score was 4, and the main concern lies in the fact that the paper does not report evaluation results on standard short-context language abilities (e.g., common-sense reasoning, in-context learning) and that the paper's pre-training configuration is slightly unusual. This was fully resolved by the authors' additional experiments during the rebuttal. With these new results, the paper is completely up to the standards of the ICML conference. Therefore, I am raising my score to 5.

**Key Questions For Authors:**

1. In Figure 2, it was mentioned that the model was fine-tuned from 2K to 65K, but the plot only shows the statistics from 512 to 16K. An explanation for this discrepancy would be helpful.
2. Can you re-run the experiments using varying model scales (both smaller and larger models would be valuable), and with varying amounts of training data? This would greatly strengthen the soundness of the empirical value of the proposed method.
3. Can you report the training and evaluation loss during pre-training and fine-tuning for all of the models? This is important for a more comprehensive understanding of the effectiveness of LUCID.
4. Can you evaluate LUCID and the baselines on common benchmarks such as common-sense reasoning tasks (such as but not limited to ARC-easy/challenge, HellaSwag, WinoGrande, BoolQ, PIQA, SocialIQA, RACE, etc.), all subsets from RULER (including the word counting, variable tracking tasks), in-context learning tasks, etc. This is also important to gain a more comprehensive understanding of the effectiveness of LUCID and the baselines.
5. Can you explain the motivation behind the subtraction by $\sqrt{d}$ in Equation 5?

**Limitations:**

yes

**Strengths And Weaknesses:**

Strengths:

- The proposed method is novel and interesting. It solves an existing problem of the standard attention mechanism with good empirical results.
- The motivation of the method is clear and rigorously explained with proofs on the properties of gradients.
- The empirical results of the proposed method are strong for retrieval-based tasks.
- The implementation of LUCID is a clever trick that avoids computing the inverse of a quadratic matrix with shape $N\times N$.


Weaknesses:

- **The motivation behind subtraction by $\sqrt{d}$**. The authors never explained the motivation behind the subtraction by $\sqrt{d}$ in Equation 5, which seems like a very important design choice.
- **Missing important evaluation results.** Currently, the authors have only reported empirical results on retrieval-related tasks, which is interesting, but insufficient. This paper lacks evaluation on conventional pre-trained models, such as zero-shot common-sense reasoning tasks (ARC-easy/challenge, HellaSwag, WinoGrande, BoolQ, PIQA, MMLU, etc.). I think reporting the loss/perplexity of each model during both the pre-training and fine-tuning processes is also important.
- **Uncommon data to model size ratio**. The proportion of model size and training tokens is uncommon. Most LLMs are trained with a ratio of 20 training tokens per model parameter (i.e., the Chinchilla law), or more training tokens than that. In contrast, this paper uses roughly 6 tokens per parameter, which is highly unconventional, and, consequently, the results might not represent the empirical results one can get when the models are trained using more common configurations.
- **Different model sizes and training data sizes**. An important property of recent LM architectures is that their performances scale predictably with model size and the amount of training data. Thus, the authors should have re-run experiments with varying model sizes and number of training tokens to investigate the scaling behavior of this novel architecture.

Minor issues:

- The $\kappa, \circ, M$ variables are never defined.
- Line 342: Why is the head dimension set to 68 instead of 64, which is much more common? Also, what learning rate scheduler was used in the experiments?

---

> ### Author Rebuttal · Authors · 2026-03-31
>
> We appreciate the review provided and have answered the questions below.
>
> > Can you explain the motivation behind the subtraction by $sqrt{d}$ in Equation 5?
>
> The subtraction by $\sqrt{d}$ is to ensure a unit-diagonal in the exp(K_norm K_norm^T/sqrt{d} - \sqrt{d}) in order to stabilize the triangular solver. Since K_norm are a result of an rmsnorm operation, the diagonals of K_norm K_norm^T would take the value d and the division with sqrt{d} would be sqrt{d}, thus a subtraction is needed to turn the diagonals to zeros before the exp(.) operation.
>
> > The $\kappa, \circ, M$ variables are never defined.
>
> $\kappa$ is the condition number of the preconditioner as mentioned in page 3 ``Why LUCID reduces attention noise in longer-contexts?." The formula is $\kappa = \sigma_{\max}(P)/\sigma_{\min}(P)$. $\circ$ is element-wise (Hadamard) product and $M$ is the 0-1 causal mask. We will make the notation more explicit in the paper.
>
> > Line 342: Why is the head dimension set to 68 instead of 64, which is much more common? Also, what learning rate scheduler was used in the experiments?
>
> The head dimension is set to 68 rather than standard 64 to ensure that all models have the same number of parameters as PaTH has W projection that introduces 13M additional parameters. We used a cosine scheduler after linear warm-up for the first 6% of the pre-training.
>
> > In Figure 2, it was mentioned that the model was fine-tuned from 2K to 65K, but the plot only shows the statistics from 512 to 16K. An explanation for this discrepancy would be helpful.
>
> Going beyond 16K for condition number was a computational bottleneck as we relied on off-the-shelf NumPy/SciPy methods to find condition number which requires O(N^3) computation, as opposed to triangular linear solve which is O(N^2).
>
> > Can you report the training and evaluation loss during pre-training and fine-tuning for all of the models? This is important for a more comprehensive understanding of the effectiveness of LUCID.
>
> The final training and fine-tuning loss values are provided below.
>
> **Training Final Loss Values**
> | Model | Loss |
> |---|---|
> | Standard | 2.5164 |
> | Diff Transformer | 2.5112 |
> | DeltaNet | 2.6059 |
> | GLA | 2.7019 |
> | GSA | 2.7040 |
> | PaTH | 2.5055 |
> | LUCID | 2.5134 |
> | LUCID PaTH | 2.5025 |
>
> **BabiLong Finetuning Final Loss Values**
> | Model | Loss |
> |---|---|
> | Standard | 1.1517 |
> | Diff Transformer | 1.2708 |
> | PaTH | 1.1684 |
> | LUCID PaTH | 0.854 |
>
> While LUCID and LUCID-PaTH achieve comparable pretraining loss to the baselines, the advantage becomes apparent during long-context finetuning — as shown in the BABILong finetuning loss, LUCID-PaTH achieves significantly lower loss (0.854) compared to Standard (1.1517) and PaTH (1.1684), indicating that LUCID's preconditioner particularly benefits learning in longer-context settings.
>
> > Can you re-run the experiments using varying model scales (both smaller and larger models would be valuable)... Can you evaluate LUCID and the baselines on common benchmarks such as common-sense reasoning tasks...
>
> We acknowledge that tasks like common-sense reasoning tasks and in-context learning tasks, and scaling analysis will provide further insights on the capabilities and limitations of LUCID. Our motivation was to do long-context retrieval well first, as this is where the theoretical advantages of key decorrelation are most directly testable. We will include scaling experiments and mentioned evaluations in the final revision.

---

> > ### Author Rebuttal · Reviewer_xRQn · 2026-04-01
> >
> > Thank you for your response. Some of my concerns have been resolved, but two important concerns remain unsolved.
> >
> > The first is evaluation on more common benchmarks (e.g., common-sense reasoning, in-context learning, etc.). The current experimental results in the paper are limited to long-context evaluations, which are interesting but not sufficient for real-world applications of the proposed method.
> >
> > Secondly, validating the effectiveness of the proposed method on varying model scales and training data scales is essential. Currently, the method is only validated on one model size. Even worse is that the token-to-parameter ratio in the paper's experiment is rather unusual, which makes one question whether the conclusion can transfer to more usual configurations in which models are trained with a much greater token-to-parameter ratio.
> >
> > If these concerns are resolved, I am willing to raise the overall recommendation score.

---

> > > ### Author Response · Authors · 2026-04-06
> > >
> > > We thank the reviewer for the constructive feedback. We conducted additional experiments to address both remaining concerns.
> > >
> > > **Concern 1: Evaluation on Common Benchmarks**
> > >
> > > We evaluate LUCID and Softmax across commonsense reasoning, reading comprehension, and in-context learning benchmarks at all three model scales, maintaining a 20x tokens-per-parameter ratio by training both LUCID and Softmax on 500M with 10B tokens, 1B with 20B tokens, and 2B with 40B tokens. Bold indicates the better result.
> > >
> > > **0-shot Benchmarks**
> > >
> > > | Benchmark | 500M LUCID | 500M Softmax | 1B LUCID | 1B Softmax | 2B LUCID | 2B Softmax |
> > > |---|---|---|---|---|---|---|
> > > | boolq (acc) | **0.6199** | 0.5963 | 0.619 | **0.622** | **0.5985** | 0.5972 |
> > > | cb (acc) | **0.4643** | 0.3929 | **0.5893** | 0.3571 | **0.4643** | 0.3393 |
> > > | multirc (acc) | 0.5708 | **0.572** | **0.5716** | 0.5703 | **0.5714** | 0.5559 |
> > > | record (f1) | **0.5754** | 0.5754 | **0.6164** | 0.6131 | **0.6564** | 0.6519 |
> > > | sglue_rte (acc) | **0.5379** | 0.5054 | 0.5162 | **0.5487** | **0.4946** | 0.4729 |
> > > | wsc (acc) | 0.3654 | **0.3846** | **0.4135** | 0.3654 | **0.6442** | 0.4135 |
> > > | tinyHellaswag (acc) | 0.228 | **0.2879** | **0.3147** | 0.2651 | **0.3784** | 0.3302 |
> > > | tinyTruthfulQA (acc) | **0.4513** | 0.4394 | 0.4168 | **0.4264** | **0.4113** | 0.3893 |
> > > | tinyWinogrande (acc) | **0.4821** | 0.4351 | 0.4413 | **0.563** | **0.4837** | 0.4522 |
> > > | ARC Challenge | **0.2509** | 0.2474 | **0.2747** | 0.2611 | **0.2944** | 0.2884 |
> > > | ARC Easy | **0.4735** | 0.4465 | **0.5084** | 0.4962 | **0.556** | 0.5299 |
> > > | mmlu | **0.2313** | 0.23 | **0.2324** | 0.2295 | 0.2443 | **0.2491** |
> > >
> > > LUCID outperforms Softmax on the majority of benchmarks across all three model scales, with an increasing edge at larger scales.
> > >
> > > **5-shot In-Context Learning (MMLU and ARC)**
> > >
> > > | Benchmark | 500M LUCID | 500M Softmax | 1B LUCID | 1B Softmax | 2B LUCID | 2B Softmax |
> > > |---|---|---|---|---|---|---|
> > > | ARC Challenge | 0.2526 | **0.2594** | **0.2875** | 0.2867 | 0.3217 | **0.3234** |
> > > | ARC Easy | **0.5072** | 0.4958 | **0.5741** | 0.5648 | **0.6376** | 0.6317 |
> > > | mmlu | 0.2559 | **0.2578** | **0.2548** | 0.2516 | **0.2606** | 0.2569 |
> > >
> > > In the 5-shot in-context learning setting, LUCID and Softmax perform comparably, with LUCID holding a slight edge at larger scales.
> > >
> > > **Concern 2: Validation Across Model and Data Scales**
> > >
> > > **Final Pre-training Loss**
> > >
> > > | Model Size | LUCID | Softmax | Delta |
> > > |---|---|---|---|
> > > | 500M | **2.5915** | 2.6111 | -0.0196 |
> > > | 1B | **2.4293** | 2.4522 | -0.0229 |
> > > | 2B | **2.2930** | 2.3206 | -0.0276 |
> > >
> > > LUCID consistently achieves lower final pre-training loss at every scale tested.
> > >
> > > We hope these results address the remaining concerns.

---

### Official Review · Reviewer_N1fm · 2026-03-13

**Soundness:** 4
**Presentation:** 4
**Significance:** 3
**Originality:** 3
**Overall Recommendation:** 5
**Confidence:** 4

**Summary:**

The authors address a trade-off of attention temperature giving either good gradient flow or de-correlated keys with sharp retrieval. Using insights from gradient descent in an RKHS, they show that using pre-conditioning in the attention mechanism can remove this trade-off and allow both desireable properties to be achieved. They give mathematical intuitions and a theorem to support this, following up with synthetic experiments demonstrating the predicted effects. Finally, they conduct a larger scale experiment against many baselines across several tasks and show that their method, LUCID, is the most effective on each.

**Compliance With Llm Reviewing Policy:**

Affirmed.

**Final Justification:**

Rebuttal addressed my main concerns, but my score was on the upper end compared to the other reviewers. Additionally, I feel to give a 6 the paper must be exceptional, and go beyond being a solid piece of research. Because of this, I maintain my score of 5.

**Key Questions For Authors:**

1. Figure 3 has a weird cut-off, what happens if you train longer? It seems plausible that Softmax might catch up quickly.
2. Could you run experiments on open-source LLMs to check the average correlation between their keys?

**Limitations:**

Only one limitation is acknowledged about the method. I think one additional (if very reasonable) limitation is that no LLM pre-training experiments are done.

**Strengths And Weaknesses:**

### Strengths
1. The paper is overall well-written with clear claims and contributions
2. Proposed method is theoretically well-motivated
3. The method is well contextualized in its relationship to other methods
4. Synthetic experiments cleanly demonstrate theoretical properties empirically
5. Method does not incur significant extra computational costs, and maintains its performance even when accounting for these costs generously
6. The method is bench-marked against a variety of baselines and across many tasks, representing a strong empirical validation of its capabilities

### Weaknesses
1. Whilst a theorem is provided to formalize the fact that LUCID does not vanish gradients when the temperature is non-extreme, no theorem is provided to properly formalize the fact that LUCID provides precise retrieval when temperature is non-extreme (although some informal math is provided to motivate this). The theoretical claims of LUCID achieving both of these desirable properties simultaneously would be greatly improved if such a theorem was provided.
2. The layout of the paper is slightly confused, with experiments spread over sections 2.2, 2.3, and 4.
3. It would strengthen the paper to have done a small pre-training run showing that the method's performance scaled to LLM training

---

> ### Author Rebuttal · Authors · 2026-03-31
>
> We thank the reviewer for their valuable suggestions and score for the paper.
>
> > Could you run experiments on open-source LLMs to check the average correlation between their keys?
>
> **Mean Key Correlation for Open-Source Models**
>
> | Model | 512 | 1024 | 2048 | 4096 | 8192 | 16384 |
> |---|---|---|---|---|---|---|
> | Qwen | 0.6569 | 0.6183 | 0.6085 | 0.6192 | 0.6064 | 0.5806 |
> | LLaMA | 0.5834 | 0.5514 | 0.5448 | 0.5346 | 0.5110 | 0.4947 |
> | Gemma | 0.5979 | 0.6359 | 0.5768 | 0.5501 | 0.5319 | 0.5397 |
>
> We used Qwen2.5-7B-Instruct, Llama-3.1-8B-Instruct, and gemma-3-12b-it and analyzed mean of correlation across keys.
>
> > It would strengthen the paper to have done a small pre-training run showing that the method's performance scaled to LLM training
>
> Our experiments section involve pretraining of models upto 1B parameters. Our work is focused on providing substantial proof-of-work for long-context performance of LUCID.
> However, we acknowledge that scaling LUCID for larger parameters would be a good future work.
>
> > Whilst a theorem is provided to formalize the fact that LUCID does not vanish gradients when the temperature is non-extreme, no theorem is provided to properly formalize the fact that LUCID provides precise retrieval...
>
> We thank the reviewer for this suggestion. We clarify that "precise retrieval" refers to LUCID's ability to improve retrieval over softmax attention through key-key decorrelation via the preconditioner, while operating at non-extreme temperatures. We will refine this language in the final version to make the distinction clearer. The key-key decorrelation analysis we provide to Reviewer 9BN1 supports this empirically, and we will explore formalizing this property in future work.
>
> > Figure 3 has a weird cut-off, what happens if you train longer? It seems plausible that Softmax might catch up quickly.
>
> Thank you for raising this point. We will extend the training in Figure 3 and include the results in the final revision. Currently, we wanted to show that gradients during phase 2 can be large for softmax attention emphasizing the learnability issue with softmax.

---

> > ### Author Rebuttal · Reviewer_N1fm · 2026-03-31
> >
> > Thank you for the clear response. I look forward to seeing the promised changes in the final paper. I will maintain my score of 5 as I feel that it is still appropriate, especially when taking into account the concerns of the other reviewers.

---

### Decision · Program_Chairs · 2026-04-30

**Decision:**

Accept (regular)

**Comment:**

*Motivation:* The conditioning of the attention weights matrix, after the exponential function and before soft-max normalization, becomes large with the prompt length.

*Contribution:* A novel attention mechanism based on preconditioning which outperforms standard attention when the sequence length is large.

*Reviews summary:* All reviewers believe that the paper is well motivated, well written, and has a significant contribution combined with a convincing research story.

*Rebuttal updates:* There were concerns about the lack of some experimental comparisons, for which the authors provided more experiments and effectively addressed the concerns.

*Conclusion:* Based on the reviews and rebuttal response, the AC votes for weak acceptance.

*AC additional comment:* There is a fundamental challenge about preconditioning for long text: there are many repeated words in long contexts, hence the word embeddings and therefore the attention weights are naturally ill-conditioned and even low-rank. I recommend the authors explain why preconditioning given repeated words or word embeddings intuitively makes sense.